# Multiarm multistage randomised controlled trial of inflammatory signal inhibitors (MATIS) for patients hospitalised with COVID-19 pneumonia during the UK pandemic

Lorna Hazell [1], Clio Pillay,[2,3] Victoria Cornelius,[1] Rachel Phillips,[1] Asad Charania,[2,3] James Wason [4] Svetlana Cherlin,[4] Sinisa Savic,[5,6] Ashley Whittington,[7] Pratap Neelakantan,[8] Paul Collini,[9] Lucy Cook,[2,3] Michelle Willicome,[2,3] Dragana Milojkovic,[2,3] Onn Min Kon [10,11] Taryn Youngstein,[3,11] Andrew Innes,[2,3] Mark Thursz,[3,12] Graham S Cooke,[3,13] Nikhil Vergis,[2,3] Nichola Cooper[2,3]

LH and CP contributed equally. NV and NC contributed equally.

For numbered affiliations see end of article.

**Correspondence to**
Prof Nichola Cooper;
n.cooper@imperial.ac.uk

## ABSTRACT

**Objectives** To determine the safety and efficacy of ruxolitinib (RUX) and fostamatinib (FOS) compared with standard of care (SOC) in patients requiring hospital admission for the treatment of COVID-19 pneumonia.

**Design** Adaptive multiarm, multistage, randomised, open-label trial (three arm, two stage).

**Setting** Five hospitals in England between October 2020 and September 2022.

**Participants** Hospitalised patients (≥18 years) with COVID-19 pneumonia defined by a modified WHO COVID-19 severity grade of 3 or 4.

**Interventions** Participants were randomly assigned 1:1:1 to receive RUX (10 mg two times per day for 7 days then 5 mg two times per day for 7 days), FOS (150 mg two times per day for 7 days then 100 mg two times per day for 7 days) or SOC.

**Main outcome measures** Primary outcome was development of severe COVID-19 pneumonia (modified WHO severity grade≥5) within 14 days of randomisation. Secondary outcomes included mortality, invasive and non-invasive ventilation, venous thromboembolism, duration of hospital stay, readmissions, inflammatory markers and serious adverse events (SAEs).

**Results** At stage 1, 181 patients were randomised, with 4 assessed as ineligible post randomisation. FOS was stopped early for futility with 16 participants (27.6%, n=58) developing severe COVID-19 pneumonia compared with 15 (25.0%, n=60) in the SOC arm (adjusted odds ratio (aOR) compared with SOC: 1.12; 95% CI 0.49 to 2.58; p=0.608). RUX progressed to stage 2 but the trial was stopped early due to slow recruitment. At the final analysis, 10 participants (16.1%, n=62) developed severe COVID-19 pneumonia in the RUX arm compared with 15 (24.6%, n=61) in the SOC arm (aOR: 0.63; 95% CI 0.25 to 1.57; p=0.161). Four (7.4%) participants in the FOS arm, none in the RUX arm and three (5.5%) in the SOC arm died within 14 days of randomisation. Infections were the most frequently reported SAE and were numerically higher in the FOS (10, 17.2%) and RUX (10, 16.1%) arms compared with SOC (7, 11.5%). Two unexpected serious adverse reactions occurred in the RUX arm only.

**Conclusions** We found no evidence that FOS was superior to SOC for the treatment of COVID-19 pneumonia in patients requiring hospital admission. Due to early stopping, the trial was underpowered to establish RUX's effect in this population. Further study is needed.

**Trial registration number** NCT04581954; EUDRA-CT: https://www.clinicaltrialsregister.eu/ctr-search/trial/2020-001750-22/GB.

## STRENGTHS AND LIMITATIONS OF THE STUDY

⇒ Participants in the Multi-Arm Trial of Inflammatory Signal Inhibitors for COVID-19 trial were broadly representative of an urban population requiring hospital admission for the treatment of COVID-19 pneumonia in the UK.

⇒ We used an efficient multiarm, multistage trial design to focus research efforts on more promising treatments in the pandemic setting.

⇒ We were able to determine that fostamatinib was unlikely to be efficacious in this patient population at an early stage of the trial.

⇒ The trial was, however, subsequently stopped early due to slowing recruitment as the pandemic receded and mass vaccination was introduced.

⇒ We were therefore unable to confirm the efficacy of ruxolitinib in this patient setting.

## INTRODUCTION

Severe COVID-19 pneumonia is characterised by respiratory and multiorgan failure in the context of marked systemic inflammation and increased thrombotic risks. Elevations in circulating inflammatory molecules are

associated with poor prognosis and, in a subset of patients, drive acute lung injury which, in the early stages of the 2020 pandemic, resulted in high rates of catastrophic multiorgan failure and death.[1]

Therapeutic interventions targeting inflammatory signalling might reduce the severity of the inflammatory response phase and associated lung damage, thereby averting respiratory failure and the need for mechanical ventilation. Data from China and Italy during the pandemic suggested that immediate resolution of symptoms could be achieved using anti-interleukin 6 (IL-6) therapy and Janus kinase (JAK) inhibitors in patients with severe disease.[2 3] The spleen tyrosine kinase (SYK) pathway was also considered a potential target; it is activated by several receptors, including C-type lectin or C-type lectin-like receptors and Fc receptors.[4] Inhibition of the SYK pathway has been shown to inhibit platelet aggregation in heparin-induced thrombocytopenia models,[5] a disease with many similar features to COVID-19.

We hypothesised that there may be an early window of opportunity to treat the COVID-19 hyperinflammatory syndrome before acute lung injury leads to organ failure. We designed a randomised controlled, multiarm multistage trial in patients hospitalised with COVID-19 pneumonia of grade 3 or 4 on the modified WHO COVID-19 Severity Scale to evaluate the effect of early intervention with two inflammatory signal inhibitors which are already licensed for use in other clinical indications: ruxolitinib (RUX) (JAK inhibitor)[6] and fostamatinib (FOS) (small molecule SYK inhibitor).[7]

## METHODS

### Trial design and participants

The Multi-Arm Trial of Inflammatory Signal Inhibitors for COVID-19 (MATIS) was an open-label multiarm, multistage randomised (1:1:1) controlled trial (RCT) of RUX and FOS for the treatment of COVID-19 pneumonia compared with routine standard of care (SOC) in patients requiring hospital admission.

The study was conducted in accordance with the recommendations for physicians involved in research on human subjects adopted by the 18th World Medical Assembly, Helsinki 1964 and later revisions.[8]

### Inclusion criteria

Eligible participants were hospitalised patients in the UK aged≥18 years with clinically suspected or laboratory confirmed SARS-CoV-2 infection, radiological change consistent with COVID-19 disease, a C reactive protein (CRP) level≥30 mg/L at any time point and COVID-19 pneumonia defined by a severity grade 3 or 4 on the modified WHO severity scale (table 1).[9] The modified scale includes an additional grade to capture clinical deterioration in patients for whom escalation in organ support is not offered. Non-English speakers were eligible to join the study; hospital translation services were provided where appropriate. Female participants of childbearing

**Table 1** Modified WHO COVID-19 Severity Scale, from the WHO R&D Blueprint

| Patient state | Descriptor | Grade |
|---|---|---|
| Uninfected | No clinical or virological evidence of infection | 0 |
| Ambulatory | No limitation of activities | 1 |
| | Limitation of activities | 2 |
| Hospitalised mild disease | Hospitalised, no oxygen therapy | 3 |
| | Oxygen by mask or nasal prongs | 4 |
| Hospitalised severe disease | $SpO_2<90\%$ on $FiO_2\geq60\%$ by face mask | 5 |
| | Non-invasive ventilation, CPAP or high-flow oxygen | 6 |
| | Intubation and mechanical ventilation | 7 |
| | Ventilation+additional organ support (vasopressors, RRT, ECMO) | 8 |
| Dead | Death | 9 |

CPAP, continuous positive airway pressure; ECMO, extracorporeal membrane oxygenation; $FiO_2$, fraction of inspired oxygen; R & D, Research and Development; RRT, renal replacement therapy; $SpO_2$, peripheral arterial blood oxygen.

potential were required to agree to abstain from sexual intercourse or use effective contraception for 42 days after the last dose of study medication. Male participants were required to abstain from sperm donation for 42 days after the last dose of study medication.

### Exclusion criteria

Patients with severity below grade 3 on the WHO COVID-19 Ordinal Scale (ie, non-hospitalised patients) were not eligible to participate in the MATIS trial. Patients were also ineligible if they had ≥grade 5 severity pneumonia or had oxygen saturation levels<90% on 60% inspired oxygen at baseline or if, in the opinion of the investigator, progression to death was inevitable within the next 24 hours, irrespective of the provision of therapy or if they required either invasive or non-invasive ventilation (NIV) including continuous positive airway pressure (CPAP) or high flow nasal oxygen at any point after hospital admission and before baseline. Patients normally on NIV such as CPAP at home were eligible to take part. Further exclusion criteria were any medical condition/history or concomitant medication that would compromise participants' safety or put the participant at significant risk or compliance with study procedures or would compromise the scientific integrity of the study, any known severe allergic reactions to the investigational agents, use of drugs within the preceding 14 days known

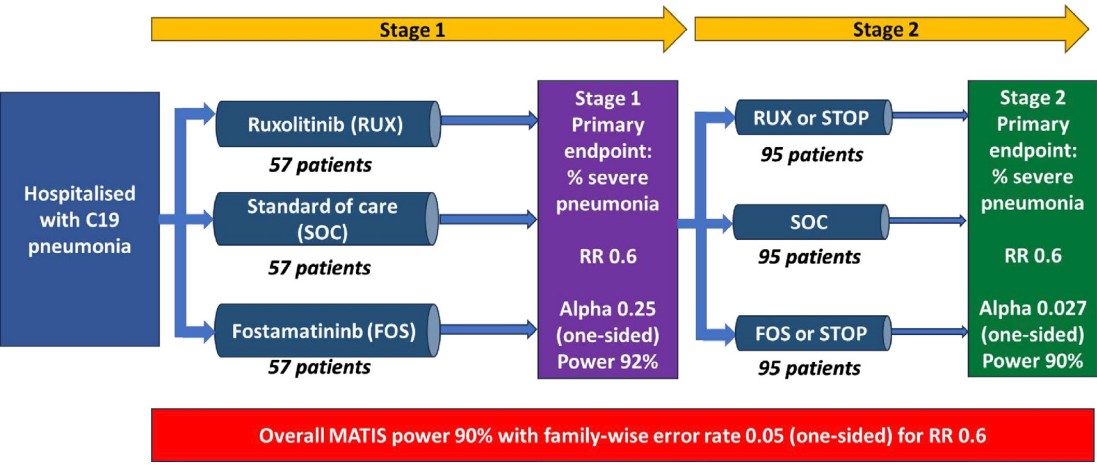

**Figure 1** Multi-Arm Trial of Inflammatory Signal Inhibitors for COVID-19 trial design.

to interact with any study treatment, Child Pugh B or C grade hepatic dysfunction, pregnancy and breast feeding.

The trial design (figure 1) is further described in the trial protocol.[10] A planned formal interim analysis (stage 1) was conducted on 10th March 2022 after 177 eligible participants had been randomised and completed 14 days of follow-up. A prespecified threshold of one-sided p value≤0.25 was required to progress treatment with RUX and/or FOS to stage 2.

### Randomisation and blinding

Participants were randomly allocated to RUX, FOS or SOC using a central web-based randomisation service that used randomisation sequences with random block sizes stratified by age (<65 vs ≥65) and study site. Staff treating participants, participant-facing research staff and participants were unblind to study treatment. Independent data monitoring committee (IDMC) reports prior to the stage 1 analysis were prepared by the trial statistician who was aware only of treatment groups labelled as group A–C. Stage 1 analysis was performed by an independent, unblind statistician at the Newcastle University. The final analyses were undertaken by an unblind trial statistician at Imperial College Clinical Trials Unit. All analyses were undertaken after the Statistical Analysis Plan (SAP) was signed off.

### Interventions

The planned duration of treatment was 14 days. Participants in the RUX arm were prescribed 10 mg two times per day for 7 days reducing to 5 mg two times per day for a further 7 days. Participants in the FOS arm were prescribed 150 mg two times per day for 7 days reducing to 100 mg two times per day for a further 7 days. Treatment was given by mouth unless the participant was unable to take oral medicines, in which case treatments were administered via a nasogastric tube. Study treatment was given in combination with SOC. Due to the evolving clinical landscape for the management of COVID-19, the SOC was not predefined and varied during the course of the trial. Coenrolment to other trials, additional treatment

and rescue therapy was permitted at the discretion of the treating clinicians.

### Follow-up

Participants were followed up at 1, 7, 14 and 28 days after randomisation. Those discharged from the hospital before day 14 completed trial medication at home and were followed up by telephone or using patient health records, where available, up to day 28.

### Outcomes

All comparisons were between RUX or FOS and SOC. The primary outcome was defined as the proportion of participants with severe COVID-19 pneumonia within 14 days of randomisation, where severe COVID-19 pneumonia was defined by a modified WHO COVID-19 Ordinal grade≥5 (on the 9-point scale, table 1).[9] This included participants who died (from any cause), required invasive ventilation or NIV (CPAP or high flow oxygen) or had $O_2$ saturation<90% on ≥60% inspired oxygen within 14 days of randomisation.

Secondary outcomes, examined at day 14 and day 28, included all-cause mortality; the proportions of participants requiring NIV (including CPAP or high flow nasal oxygen), invasive ventilation and renal replacement therapy (RRT); incidence of venous thromboembolism (VTE); length of hospital stay; change in absolute severity on the modified WHO COVID-19 Ordinal Scale (grade 3 or 4 to 5, 6, 7, 8 and 9); inflammatory markers (CRP, lactose dehydrogenase, ferritin and D-dimer); and serum creatinine levels. Safety outcomes included serious adverse events (SAEs), serious adverse reactions (SARs) and treatment discontinuations. SAEs classified as possibly, probably or definitely related to study treatment are reported in this manuscript as SARs. Due to the trial setting, the focus of expedited reporting of unexpected SARs (USARs) was on events that were highly likely to be related to study treatment only. Anticipated events were exempted from expedited reporting if they were efficacy endpoints, a consequence of COVID-19 or common in the study population. Non-SAEs were not collected.

Additional outcomes, including readmissions to hospital within 28 days, the proportion of participants with severe pneumonia at day 28 and separate components of the primary outcome were not specified as such in the trial protocol but were included in the SAP.

A protocol amendment on 2 November 2022 included the addition of a substudy to collect further data to evaluate longer-term outcomes. The results from this substudy will be published separately from the current manuscript.

### Sample size

A sample size of 171 (57 per arm) participants at stage 1 and an additional maximum of 285 (95 per arm) was selected to provide power of 90% (minimum marginal power) with a maximum 5% chance of an intervention arm being recommended when it provides no improvement over control (5% one-sided family-wise error rate), inflated by 5% for missing outcome data (full details are described in the trial protocol).[10] This sample size calculation assumed a 50% rate of severe pneumonia in the SOC arm and a reduction of this risk in an experimental arm to 30% (relative risk, 0.6).

### Statistical analysis

Participant flow through the trial was described using a Consolidated Standards of Reporting Trials flow chart.[11] A scatter plot was used to show the trajectory of participant recruitment over time. Simple descriptive statistics were used to summarise baseline participant characteristics, co-enrolment to other trials and SOC medications received at baseline and post randomisation.

The modified intention-to-treat (ITT) population included all eligible randomised participants analysed in the treatment arm to which they were allocated, regardless of treatment subsequently received. Participants found to have been randomised in error were excluded from the modified ITT population. Participants were assumed to have received 14 days of treatment unless there was evidence for stopping earlier.

For the primary outcome, we used a generalised linear model with a binomial distribution and logit link function to compare the odds of developing severe COVID-19 pneumonia between RUX or FOS and SOC. At stage 1, two separate models were used to compare the effects of FOS versus SOC and RUX versus SOC. These models were adjusted for using a propensity score derived from a logistic regression model that included baseline modified WHO COVID-19 severity grade (3 or 4), age (<65 or ≥65), receipt of steroids at baseline and receipt of IL-6 inhibitors at baseline.

For the final analysis, we included all three trial arms in a single model. We planned to adjust our final analysis model for site, age category (<65 or ≥65), baseline modified WHO COVID-19 severity grade (3 or 4) and receipt of treatments assumed to be effective prior to randomisation based on current knowledge at that time of the MATIS trial. These effective treatments included dexamethasone, IL-6 inhibitors, selected antivirals and COVID vaccination. As the efficacy of remdesivir was unconfirmed at the time of the trial, Paxlovid® and/or molnupiravir were the only antivirals specified a priori; however, no participants received these agents. As a large proportion of participants had received dexamethasone in each arm, prior steroid use was not included in our final analysis model to avoid instability due to zero cell counts in some strata. Similarly, as there were several sites with very few randomised participants, we excluded study site from the primary analysis model. Thus, our final model adjusted for prior use of IL-6 inhibitors, age category, modified WHO COVID-19 severity grade and receipt of COVID-19 vaccine. These were included as separate covariates in the final model. Multiple imputation was used to account for any missing data such that all eligible randomised participants were included in both the stage 1 and final analyses (see online supplemental file 5).[12] Treatment effects were reported as ORs with corresponding 95% CIs and one-sided p values.

The original two-stage design specified that significance would be judged at the 0.027 level (ie, a two-sided p value<0.054 was to be used to indicate a significant treatment effect). However, as the planned sample size was not reached, the p values presented should not be interpreted with reference to this threshold as the trial was significantly underpowered. Interpretation should be focused on the point estimate and 95% CI as a way to assess the evidence of a treatment effect and associated uncertainty; however, this interpretation should be regarded as hypothesis-generating only.

Analysis of secondary outcomes was performed for the final analysis only. Here, the primary analysis model was repeated to compare the odds of developing severe pneumonia by day 28. This model was also run separately for each individual component of the primary outcome (death, NIV, invasive ventilation with and without organ support and $O_2$<90% on ≥60% inspired oxygen) and for the maximum COVID-19 severity grade reached by day 14 and day 28. The trial protocol included a secondary objective to compare the proportions of participants requiring invasive ventilation and/or extracorporeal membrane oxygenation (ECMO); however, this was not assessed as no participants received ECMO in MATIS. COVID-19 severity grade was further modelled as an ordinal variable using a mixed ordinal logistic regression model to compare the odds of progression to more severe disease at day 14. Mixed linear regression models were used to compare changes in inflammatory markers and serum creatinine up to day 28, and a Cox proportional hazards model was used to compare the rates of hospital discharge between trial arms. The frequencies of RRT, VTE, readmission within 28 days, SAEs, SARs and treatment discontinuations were summarised descriptively. Dot plots were used to visually display between-group differences in SAE frequencies (as risk ratios), grouping events at the system organ class level.

Prespecified sensitivity analyses were undertaken to examine alternative approaches to handling missing data

first, assuming that all participants with missing outcome had poor outcome eg, all missing assumed severity grade≥5, and second assuming that all participants with missing outcome had good outcome, for example, all missing assumed severity grade<5. A planned sensitivity analysis of the primary outcome, adjusting for receipt of effective COVID-19 treatments through coenrolment to the active arm of another RCTwas not performed since such treatments were not found to have been prescribed to trial participants as part of coenrolment to another trial. Another sensitivity analysis to explore both the direct effect of treatment on the outcome and any indirect effect of receiving known effective treatments as rescue medications was not conducted, as all of the participants who received these medications post randomisation to MATIS received the medication on or after the index date for the primary outcome.

We performed two prespecified exploratory analyses to assess the impact of adherence to treatment on the primary outcome. First, we conducted a naïve per-protocol analysis in participants who had a minimum of 90% of the planned treatment (at least 13 days of active treatment). In the SOC arm, patients were deemed to be 100% adherent to SOC by definition. Patients who died were considered adherent if they had not discontinued trial treatment at the time of death. Second, we calculated the treatment effect (as a risk difference (RD)) among compliers using a complier-adjusted case (CACE) analysis with a two-stage residual inclusion estimator approach[13] and randomisation group as the instrumental variable. Estimates from the CACE analysis were generated from two separate models for each treatment comparison, while the original ITT analysis was one model.

Additional exploratory analyses were performed to assess the impact of the time of enrolment to the trial. A categorical variable was created by splitting the recruitment period into quartiles such that participants were grouped into approximately four equal-sized groups based on time of enrolment into the MATIS study. This variable was included as a covariate in the primary analysis model as a treatment group-by-time interaction term to assess the impact of time of recruitment on the treatment effect estimate. A crude OR was calculated for each quartile. This analysis was performed using the unimputed dataset since the multiple imputation model could not produce the required estimates due to inconsistency in the omitted variables across the different imputed datasets. Subgroup analysis of the primary outcome by age, sex, body mass index, cardiovascular disease, chronic lung disease, end-stage renal failure, liver cirrhosis, diabetes, baseline CRP and D-dimer, immunocompromisation, smoking status and number of days since onset of COVID-19 at randomisation was also performed.

Stage 1 analysis was performed in R (https://www.R-project.org/). All other analysis was undertaken using Stata/IC V.16.1 (StataCorp, College Station, Texas, USA).

The SAP is provided in online supplemental appendix 1.

### Patient and public involvement

Information on MATIS trial design was included on clinicaltrial.gov. An X page was set up to explain the trial to a wider audience, and the protocol was published in Trials (2021).[10]

## RESULTS

### Participant flow and baseline characteristics

Between October 2020 and September 2022, 185 patients from five UK hospitals in England were enrolled (figure 2).

The majority of participants were recruited from one London teaching hospital site (n=170) with a smaller proportion from other UK city hospitals (n=15). Recruitment followed the peaks and troughs of the COVID-19 pandemic waves observed in the UK during the trial period (figure 3). This period predominantly covered the Alpha and Delta waves in the UK.[14] Although the Omicron variant was detected from around December 2021, only 12 participants were recruited to MATIS after that date.

During the 28-day follow-up, 13 participants withdrew consent and one was lost to follow-up. Four patients found to be ineligible post randomisation were excluded from the analyses. A slightly higher proportion of participants completed follow-up but discontinued study medication early in the FOS arm (13, 22.4%) than in the RUX arm (11, 17.2%). 38 (66.7%) participants in the FOS arm and 46 (71.9%) in the RUX arm were assumed to have completed the 14-day treatment course.

A higher proportion of participants in the RUX arm compared with the SOC arm were female (35.9% vs 30.2%), were of white ethnicity (48.4% vs 42.9%), had a modified WHO COVID-19 severity grade of 3 at baseline (21.9% vs 14.3%) or had received a COVID-19 vaccination (25.0% vs 17.5%) (table 2). In contrast, these characteristics were less prevalent in the FOS arm compared with SOC, with the exception of the baseline severity grade of 3 (17.2% in FOS vs 13.3% in SOC). The percentage of participants with a baseline modified WHO COVID-19 severity grade of 4 decreased over the trial recruitment period from 84.8% in the Alpha wave to 75% for those recruited in the Omicron wave, while the percentage of participants who had received the COVID-19 vaccination increased from 8% to 50% (online supplemental table S1).

The majority of participants in all three trial arms were receiving anticoagulant therapy at baseline (table 3), most were receiving an immuno-modulating agent, specifically dexamethasone (online supplemental table S2). Just under half of participants had received an antiviral agent (table 3), of whom 86 (98.9%) received remdesivir. These SOC treatments were initiated a median of 1 day before or on the same day as randomisation in all three trial arms and continued once randomised with similar durations in all three arms ranging from a median of 5 days for antivirals and 9 days for anticoagulants (table 3). More

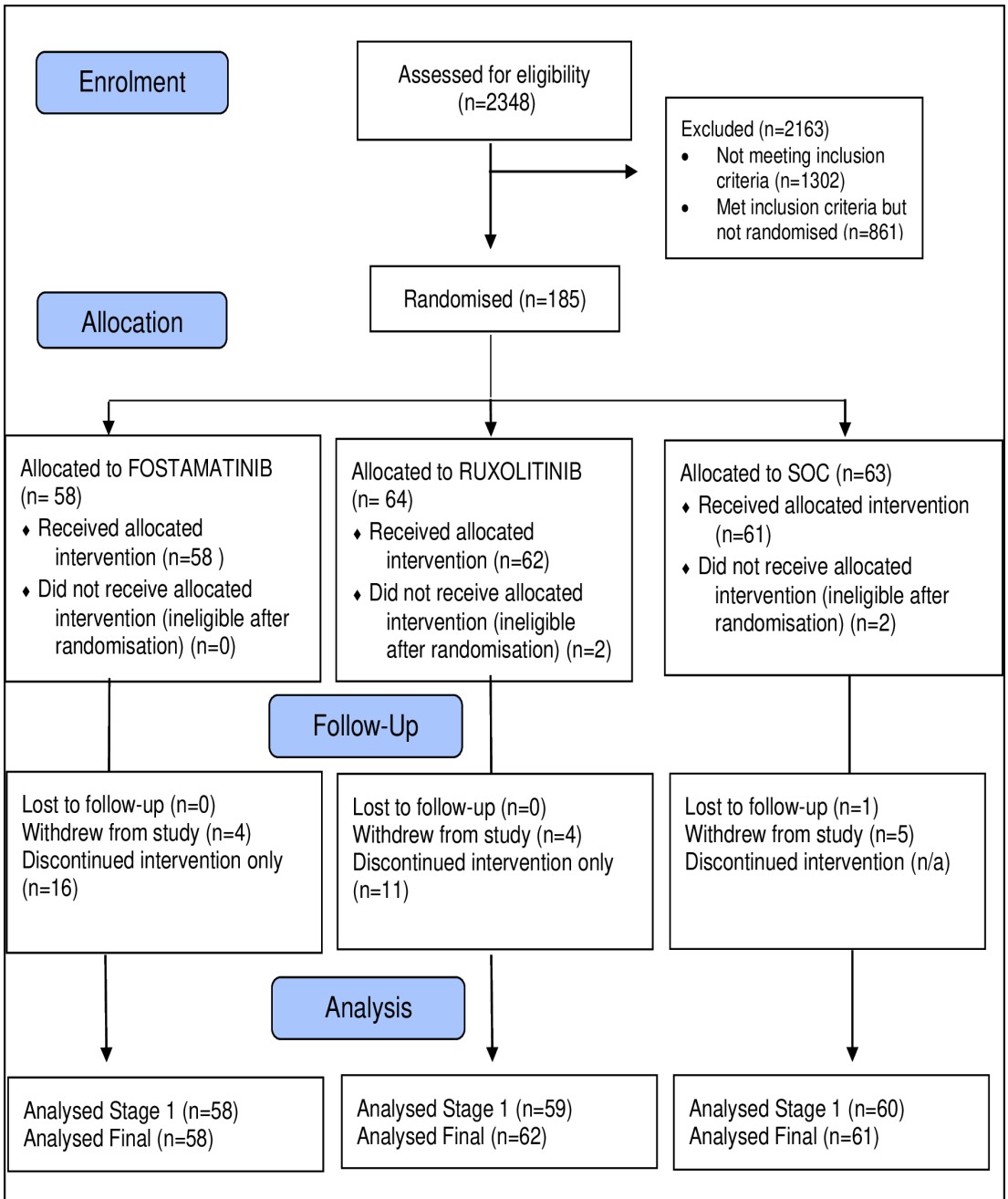

**Figure 2** Participant flow. SOC, standard of care.

participants in the SOC arm had received antibiotics at baseline (74.6%) compared with the FOS (58.6%) or RUX (64.1%) arms, whereas fewer participants initiated antibiotics after randomisation in the SOC arm (22.2%) compared with the FOS (32.8%) and RUX (25.0%) arms (online supplemental table S3). While fewer participants in the SOC (14.3%) arm had received an IL-6 inhibitor at baseline compared with the FOS (19.0%) and RUX (20.3%) arms (online supplemental table S2), these were initiated more frequently after randomisation in the SOC arm (14.3%) compared with the FOS (6.9%) and RUX (9.4%) arms (online supplemental table S4).

Prior to randomisation, similar proportions (approximately 6–9%) of participants had been coenrolled on another RCT across the three trial arms. Coenrolment after randomisation to MATIS was higher in the SOC arm (11.1%) and RUX arm (9.4%) than in the FOS arm (3.4%) (online supplemental table S5).

### Primary outcome

At stage 1 (n=177), there was no evidence of a reduction in the odds of progression to severe COVID-19 pneumonia for FOS: adjusted OR, compared with SOC: 1.12; 95% CI 0.49 to 2.58; one-sided p value=0.608 (table 4). The IDMC recommended stopping the FOS arm for futility at this stage on the basis of the one-sided p value exceeding the prespecified threshold of 0.25.[10] Recruitment to the RUX (and SOC) arm was recommended to continue (p=0.152,

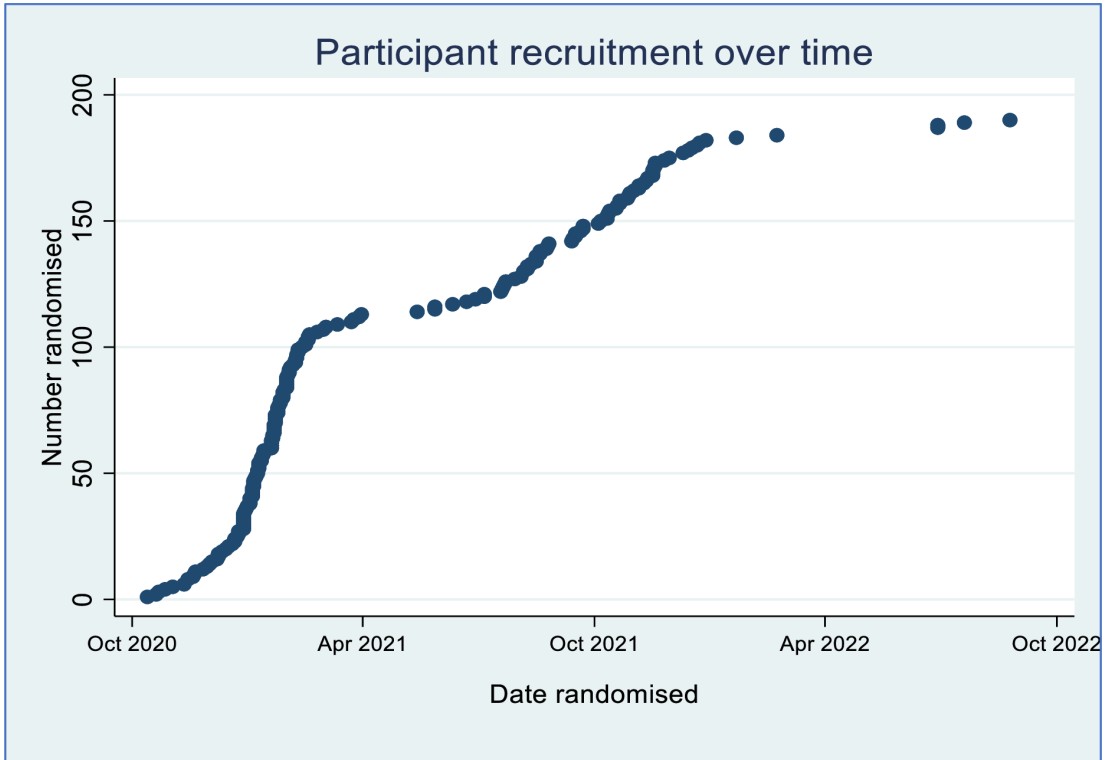

**Figure 3** Recruitment over time.

table 4); however, due to the slow recruitment because of the reduction in new COVID-19 cases, the trial was stopped early on 2 September 2022 after only recruiting an additional four participants. Therefore, the final analysis presented in this manuscript includes all three trial arms as if they had been a conventional one-stage RCT comparing RUX and FOS with SOC.

In the final analysis (n=181, table 4), the numbers (%) of participants with severe COVID-19 pneumonia by day 14 were 16 (27.6%, n=58) and 10 (16.1%, n=62) in the FOS and RUX arms, respectively, and 15 (24.6%, n=61) in the SOC arm. There remained a numerically higher odds of reaching a modified WHO COVID-19 severity grade≥5 in the FOS arm compared with SOC (OR: 1.19; 95% CI 0.51 to 2.76; nominal one-sided p value=0.659). In contrast, a numerically lower odds of severe COVID-19 pneumonia was observed for RUX compared with SOC (OR: 0.63; 95% CI 0.25 to 1.57; nominal p=0.161). In both comparisons, the 95% CI was wide and included 1. Study conclusions remained unchanged when applying alternative missing data assumptions (online supplemental table S6).

## Secondary outcomes

Results from the primary analysis model when extended to include data up to day 28 (modified WHO COVID-19 severity grade≥5 by day 28) were very similar to that for day 14 (online supplemental table S7). The primary outcome at day 14 and day 28 was most commonly triggered by participants experiencing oxygen saturation levels<90% on receipt of≥60% inspired oxygen or receipt

of NIV (online supplemental tables S8–S10). Seven participants died (four in the FOS arm and three in the SOC arm) on or before day 14 and a further three died (two in the RUX arm and one in the FOS arm) on or before day 28. Individual models for each of the components of the primary outcome are presented; however, event counts are small and resulting model estimates are imprecise. Results from the mixed ordinal logistic regression model were broadly consistent with the primary analysis model (online supplemental table S11).

Nine participants (four in the FOS arm, three in the RUX arm and two in the SOC arm) received mechanical ventilation by day 14, of whom six received organ support (inotropes only; no participants received ECMO) (online supplemental file 8). Separately, one participant with pre-existing ESRF received RRT. Six participants (two in the FOS arm, three in the RUX arm and one in the SOC arm) had a VTE event including one participant with a serious VTE in the SOC arm. Two participants (one in the RUX and one in the FOS arm) had the VTE event on the day of randomisation.

There was no evidence of a difference in the time to discharge from randomisation for either FOS or RUX compared with SOC (online supplemental figure S1). HRs from the Cox model were 0.96 (95% CI 0.64 to 1.43) for FOS versus SOC and 1.05 (95% CI 0.71 to 1.55) for RUX versus SOC, after adjusting for age, site, prior COVID-19 vaccination, receipt of steroids and receipt of IL-6 inhibitors at baseline. Readmission by day 28 was more frequent in the RUX (eight participants, 13.8%)

**Table 2** Baseline characteristics by treatment arm

| Patient characteristic, n (%) unless otherwise specified | Fostamatinib (n=58) | Ruxolitinib (n=64) | Standard of care (n=63) |
|---|---|---|---|
| Age in years | | | |
| Mean (SD) | 58.9 (15.0) | 60.5 (15.8) | 59.1 (16.3) |
| Sex | | | |
| Male | 43 (74.1) | 41 (64.1) | 44 (69.8) |
| Female | 15 (25.9) | 23 (35.9) | 19 (30.2) |
| Ethnicity | | | |
| White | 21 (36.2) | 31 (48.4) | 27 (42.9) |
| Mixed or multiple ethnic groups | 1 (1.7) | 0 (0.0) | 1 (1.6) |
| Asian or Asian British | 7 (12.1) | 11 (17.2) | 8 (12.7) |
| Black, black British, Caribbean or African | 5 (8.6) | 7 (10.9) | 7 (11.1) |
| Other ethnic group | 24 (41.4) | 15 (23.4) | 20 (31.7) |
| Body mass index in kg/m$^2$ | | | |
| N (N missing) | 55 (3) | 57 (7) | 60 (3) |
| Mean (SD) | 30.1 (6.5) | 30.6 (7.1) | 29.9 (6.9) |
| Modified WHO COVID-19 severity | | | |
| Grade 3 | 10 (17.2) | 14 (21.9) | 9 (14.3) |
| Grade 4 | 48 (82.8) | 50 (78.1) | 54 (85.7) |
| Time from onset of symptoms | | | |
| N (N missing) | 57 (1) | 51 (13) | 57 (6) |
| Mean (SD) | 10 (4) | 9 (4) | 10 (4) |
| Chronic lung disease | | | |
| N (N missing) | 58 (0) | 62 (2) | 63 (0) |
| Yes | 7 (12.1) | 12 (19.4) | 11 (17.5) |
| Diabetes | | | |
| N (N missing) | 58 (0) | 62 (2) | 63 (0) |
| Yes | 15 (25.9) | 19 (30.6) | 20 (31.7) |
| Hypertension | | | |
| N (N missing) | 58 (0) | 62 (2) | 63 (0) |
| Yes | 25 (43.1) | 28 (45.2) | 28 (44.4) |
| Ischaemic heart disease | | | |
| N (N missing) | 58 (0) | 62 (2) | 63 (0) |
| Yes | 11 (19.0) | 10 (16.1) | 10 (15.9) |
| Heart failure | | | |
| N (N missing) | 58 (0) | 62 (2) | 63 (0) |
| Yes | 2 (3.4) | 3 (4.8) | 2 (3.2) |
| Immunocompromised | | | |
| N (N missing) | 58 (0) | 62 (2) | 63 (0) |
| Yes | 1 (1.7) | 5 (8.1) | 2 (3.2) |
| End-stage renal failure | | | |
| N (N missing) | 58 (0) | 62 (2) | 63 (0) |
| Yes | 4 (6.9) | 6 (9.7) | 3 (4.8) |
| Liver cirrhosis | | | |
| N (N missing) | 58 (0) | 62 (2) | 63 (0) |
| Yes | 0 (0.0) | 0 (0.0) | 0 (0.0) |

Continued

**Table 2** Continued

| Patient characteristic, n (%) unless otherwise specified | Fostamatinib (n=58) | Ruxolitinib (n=64) | Standard of care (n=63) |
|---|---|---|---|
| Current smoker | | | |
| N (N missing) | 58 (0) | 62 (2) | 62 (1) |
| Yes | 0 (0.0) | 2 (3.2) | 3 (4.8) |
| Prior COVID-19 vaccination | | | |
| N (N missing) | 58 (0) | 64 (0) | 63 (0) |
| Yes | 8 (13.8) | 16 (25.0) | 11 (17.5) |
| Serum creatinine (µmol/L) | | | |
| N (N missing) | 57 (1) | 62 (2) | 63 (0) |
| Mean (SD) | 133 (203) | 119 (145) | 119 (172) |
| C reactive protein (mg/L) | | | |
| N (N missing) | 57 (1) | 62 (2) | 63 (0) |
| Mean (SD) | 94.1 (61.5) | 120.9 (78.4) | 105.8 (69.5) |
| Lactate dehydrogenase (IU/L) | | | |
| N (N missing) | 36 (22) | 37 (27) | 36 (27) |
| Mean (SD) | 419 (208) | 488 (309) | 439 (136) |
| Ferritin (µg/L) | | | |
| N (N missing) | 52 (6) | 52 (12) | 47 (16) |
| Mean (SD) | 1471 (1459) | 1671 (3069) | 1254 (778) |
| D-dimer (ng/mL) | | | |
| N (N missing) | 55 (3) | 53 (11) | 53 (10) |
| Mean (SD) | 1601 (2749) | 1732 (2614) | 1341 (2781) |

and SOC (8, 14.6%) arms than in the FOS arm (3, 5.6%) (online supplemental table S12). No difference in levels of inflammatory markers was observed between trial arms (online supplemental table S13). Statistically significant differences in serum creatinine levels were noted with higher mean values in the FOS arm at day 14 (adjusted mean difference vs SOC: 34.3 µmol/L; 95% CI 6.8 to 61.8) and at day 28 (43.6 µmol/L; 95% CI 15.7 to 71.5) and in the FOS arm at day 28 (adjusted mean difference vs SOC: 27.2 µmol/L; 95% CI 0.6 to 53.7) (online supplemental table S13).

### Safety

Safety was evaluated in all 181 eligible, randomised participants. Four patients were excluded from this analysis as they were randomised in error. A total of 48 SAEs were reported in 38 participants with a higher proportion experiencing at least one SAE in the RUX (14, 22.6% of participants) and FOS arms (13, 22.4%) compared with SOC (11, 17.5%) (table 5).

Three of the reported SAEs were considered to be possibly related to study treatment. These are reported here as SARs and included one expected SAR in the FOS arm (lung infection) and two unexpected SARs in the RUX arm (one case each of hypotension and lymphoedema). The most frequently reported SAE in all three trial arms was infections with numerically higher frequencies in the

FOS (10, 17.2%) and RUX (10, 16.1%) arms compared with SOC (7, 11.5%) (online supplemental figures S2 and S3). These included five cases of lung infection in the RUX arm (four participants) and three cases in the FOS arm (three participants), both higher than the SOC arm (one participant). One event of lung infection in the RUX arm resulted in a fatal outcome. None of the 10 fatal events recorded in the trial was considered to be related to study treatment.

### Exploratory analyses

Results from the per-protocol analysis in participants who received ≥90% of the prescribed dose were consistent with the primary analysis (online supplemental table S14). The complier-adjusted RD estimates were slightly further away from the null than the corresponding ITT analyses but did not reach statistical significance (complier adjusted vs ITT: FOS vs SOC: RD 0.05 vs 0.03; RUX vs SOC: RD −0.10 vs −0.06). Stratification of the analysis by grouping participants into approximately four equal-sized groups based on time of randomisation resulted in somewhat imprecise estimates and was inconclusive (online supplemental table S15). There was no strong evidence for any subgroup effects (online supplemental figures S4 and S5). These results should be interpreted with caution due to the small sample size within subgroups.

**Table 3** SOC treatments received at or prior to randomisation by treatment arm

| Drug group | FOS (n=58) | RUX (n=64) | SOC (n=63) |
|---|---|---|---|
| Antiviral agents | | | |
| Received≥1 dose, n (%) | 27 (46.6) | 31 (48.4) | 29 (46.0) |
| Median (IQR) time since initiation (days); n missing | 0 (0, 1); 0 | 0 (0, 1); 0 | 0 (0, 1); 0 |
| Median (IQR) duration (days); n missing | 5 (4, 6); 0 | 5 (5, 6); 0 | 5 (4, 5); 0 |
| Immuno-modulating agents | | | |
| Received≥1 dose, n (%) | 56 (96.6) | 58 (90.6) | 56 (88.9) |
| Median (IQR) time since initiation (days); n missing | 1 (0, 1); 0 | 1 (0, 1); 0 | 1 (0, 1); 0 |
| Median (IQR) duration (days); n missing | 6 (4, 9); 0 | 8 (5, 11); 0 | 7 (4, 10); 1 |
| Antibiotics | | | |
| Received≥1 dose, n (%) | 34 (58.6) | 41 (64.1) | 47 (74.6) |
| Median (IQR) time since initiation (days); n missing | 1 (0, 2); 0 | 1 (0, 2); 0 | 1 (1, 2); 0 |
| Median (IQR) duration (days); n missing | 5 (2, 7); 0 | 4 (2, 7); 0 | 6 (2, 7); 0 |
| Anticoagulation treatment | | | |
| Received≥1 dose, n (%) | 54 (93.1) | 54 (84.4) | 57 (90.5) |
| Median (IQR) time since initiation (days); n missing | 1 (0, 1); 0 | 1 (0, 1); 0 | 1 (0, 1); 0 |
| Median (IQR) duration (days); n missing | 9 (5, 25); 1 | 9 (5, 13); 1 | 9 (6, 13); 0 |
| Antiplatelet therapy | | | |
| Received≥1 dose, n (%) | 6 (10.3) | 1 (1.6) | 2 (3.2) |
| Median (IQR) time since initiation (days); n missing | 1 (0, 1); 0 | 1 (1, 1); 0 | 2 (0, 2); 0 |
| Median (IQR) duration (days); n missing | 17 (3, 30); 4 | Missing | 1 (1, 1); 0 |
| Convalescent serum therapy | | | |
| Received≥1 dose, n (%) | 0 (0.0) | 1 (1.6) | 1 (1.6) |
| Median (IQR) time since initiation (days); n missing | N/A | 6 (6, 6); 0 | 0 (0, 0); 0 |
| Median (IQR) duration (days); n missing | N/A | 1 (1, 1); 0 | 2 (2, 2); 0 |
| Ronapreve | | | |
| Received≥1 dose, n (%) | 1 (1.7) | 1 (1.6) | 0 (0.0) |
| Median (IQR) time since initiation (days); n missing | 0 (0, 0); 0 | 0 (0, 0); 0 | N/A |
| Median (IQR) duration (days); n missing | 1 (1, 1); 0 | 1 (1, 1); 0 | N/A |

FOS, fostamatinib; N/A, not applicable; RUX, ruxolitinib; SOC, standard of care.

## DISCUSSION
### Principal findings

MATIS was an open-label, multicentre, multiarm and multistage RCT of RUX and FOS for COVID-19 pneumonia compared with routine SOC. The study was designed between March and April 2020, with participants recruited largely from the second and third waves of the UK pandemic, with the majority of participants recruited from three hospitals in North-West London. Our efficient, multistage, multiarm design allowed early stopping of the FOS arm for futility. This design allows recruitment efforts to focus on more promising treatments, a particularly important feature in the pandemic setting. However, due to the success of the COVID-19 vaccine leading to a dramatic reduction in hospitalisation and mortality, the study was closed in September 2022, before reaching the power needed to determine the effect of ruxolitinib in the treatment of COVID-19 pneumonia in patients requiring hospital admission.

Although the study was not powered to assess the secondary outcome of death, no deaths were recorded in the RUX arm in the first 14 days while four participants in the FOS arm and three in the SOC arm died during this period. There was no difference in time to discharge between active treatment and SOC, but there were fewer readmissions in the FOS arm. In terms of safety, there were higher numbers of lung infections in the active treatment arms, including one infection-related death in the RUX arm. However, this event was not considered to be related to study treatment. We noted an increase in serum creatinine in both arms; however, the clinical relevance of this finding is unclear.

Approximately one-third of participants did not complete the 14-day treatment course for FOS and RUX.

**Table 4** Primary outcome statistical model results

| | | | Unadjusted | | Adjusted† | |
|---|---|---|---|---|---|---|
| Grade≥5 by day 14 | N | n* (%) | OR (95% CI) | P value‡ | OR (95% CI) | P value‡ |
| Stage 1 | | | | | | |
| Fostamatinib | 58 | 16 (27.6) | 1.13 (0.50 to 2.58) | 0.766 | 1.12 (0.49 to 2.58) | **0.608** |
| Ruxolitinib | 59 | 10 (16.9) | 0.62 (0.25 to 1.53) | 0.300 | 0.60 (0.22 to 1.60) | **0.152** |
| SOC | 60 | 15 (25.0) | 1.0 | | 1.0 | |
| Final analysis | | | | | | |
| Fostamatinib | 58 | 16 (27.6) | 1.19 (0.52 to 2.71) | 0.684 | 1.19 (0.51 to 2.76) | **0.659** |
| Ruxolitinib | 62 | 10 (16.1) | 0.59 (0.24 to 1.44) | 0.243 | 0.63 (0.25 to 1.57) | **0.161** |
| SOC | 61 | 15 (24.6) | 1.0 | | 1.0 | |

*Multiple imputation used at stage 1 and final analysis to impute primary outcome status for three patients in the FOS arm, four in the RUX arm and four in the SOC arm.
†Stage1: propensity score adjusted; final analysis: adjusted for baseline modified WHO COVID-19 severity, age category, use of IL-6 inhibitor and prior COVID-19 vaccination.
‡P values are two-sided except for those in bold where one-sided p values are presented; p values for final analysis are nominal as planned sample size not reached.
FOS, fostamatinib; IL-6, interleukin 6; RUX, ruxolitinib; SOC, standard of care.

The results of our CACE analyses, which estimated the impact of adherence on the treatment effects, were consistent with our primary analysis. However, due to the pandemic setting, adherence measures were based on self-report for participants given take-home medication following hospital discharge rather than in-person pill counts. Our estimate of adherence and its impact on treatment effects may therefore be underestimated.

**Comparison with other studies**

RUX is an inhibitor of the JAK/STAT pathway (Janus kinases and Signal Transducer and Activator of Transcription proteins), specifically inhibiting subtypes JAK1 and JAK2. It has a rapid mode of action with cytokine-induced STAT3 phosphorylation being inhibited within 2 hours of administration, resulting in a reduction of tumour necrosis factor-alpha, IL-6 and CRP. Several other trials studied the effect of RUX during the pandemic, with variable responses reported, some positive, but most were neutral.[15–17] There were many differences between studies including: types of patients recruited (ie, differing severity of COVID-19), timing of the study in the pandemic (resulting in variability of the virus over time, changes in SOC; and the vaccine roll-out in 2021) and different primary outcomes. As with MATIS, some of these trials were underpowered due to early termination. Furthermore, the appropriate dosing of RUX to treat or prevent hyperinflammation is yet to be determined. We chose a low to intermediate dose for MATIS, because we had concerns regarding immunosuppression during the active phase of the infection. This dose appeared to be appropriate for this cohort of patients given the lower rate of progression to more severe disease and absence of deaths before 14 days in the RUX arm. Higher doses such as 25 mg once a day, used in conditions such as graft versus host disease,[18] or longer duration of treatment (>14 days) may be more appropriate for patients with more severe COVID-19.

Given the CRP and IL-6 drive in COVID-19, several clinical trials of other agents in this class (including baracitinib and tofacitinib) targeting patients at different disease stages were performed.[19 20] A meta-analysis of four RCTs of baracitinib in hospitalised patients with

**Table 5** Summary of serious adverse events type by treatment arm

| | Fostamatinib (n=58) | | Ruxolitinib (n=62) | | Standard of care (n=61) | |
|---|---|---|---|---|---|---|
| Event type | Events (n) | Participants with events, n (%) | Events (n) | Participants with events, n (%) | Events (n) | Participants with events, n (%) |
| SAE | 14 | 13 (22.4) | 18 | 14 (22.6) | 16 | 11 (17.5) |
| SAR | 1 | 1 (1.7) | 2 | 2 (3.2) | 0 | 0 (0.0) |
| USAR | 0 | 0 (0.0) | 2* | 2 (3.2) | 0 | 0 (0.0) |

*This included one patient with hypotension and one patient with lymphoedema.
SAE, serious adverse event; SAR, serious adverse reaction; USAR, unexpected serious adverse reaction.

COVID-19, including 10 815 patients, showed a reduction in 28-day mortality (OR 0.69, 95% CI 0.5 to 0.94; p=0.04).[21] Results from two of the large studies included in the meta-analysis, The Randomised Evaluation of COVID-19 Therapy (RECOVERY) and COV-BARRIER studies,[22 23] led to the US Food and Drug Administration allowing the emergency authorisation of baracitinib with remdesivir in patients with COVID-19 requiring supplemental oxygen, invasive mechanical ventilation or ECMO.[24]

The RECOVERY study also observed a 15% reduction in mortality for tocilizumab (an IL-6 inhibitor) compared with usual care.[25] Tociluzimab was added to SOC during MATIS and was used in approximately one-fifth of the patients in this trial.

FOS is an SYK pathway inhibitor, licensed for use in patients with immune thrombocytopenia. The SYK pathway appears to be critical in the pathogenesis of COVID-19 as well as other SARS infections, modulating cytokine secretion and promoting the influx of immune cells such as macrophages and monocytes into the lung, contributing to both the hyperinflammatory state and long-term lung damage. As such, SYK inhibition appeared as a novel mechanism to treat COVID-19 pneumonia through the prevention of cytokine release, lung inflammation and platelet activation, all critical events in COVID-19 pathway.[26 27]

Despite a plausible biological mechanism of action, we found no evidence that FOS prevented progression of COVID-19 to more severe disease in this cohort of patients. It is possible that our patient cohort was too mildly affected to have any benefit from FOS. It is also possible we did not use the correct dose, or duration of FOS. Another RCT comparing FOS 100 mg two times per day plus SOC to placebo plus SOC included patients with more severe COVID-19 (over half of the patients required NIV or high-flow oxygen devices (56.7% FOS and 51.7% placebo), or invasive mechanical ventilation and/or ECMO (36.7% FOS and 41.1% placebo)).[28] In that study, the rates of SAEs by day 29 (the primary end point) were not statistically different between the FOS group and the placebo group (10.5% vs 22.0%, p=0.2), while secondary endpoints, which the study was not powered for, such as 28-day mortality, days on supplemental oxygen and ordinal scores of clinical status favoured FOS. Both the inclusion criteria and the dose of FOS were different to MATIS, limiting their comparison, but potentially implying the timing of the use of FOS in diseases such as COVID-19 might be critical.

Another double-blind, randomised, placebo-controlled trial of FOS in patients with COVID-19 on supplementary oxygen (the FOCUS trial) recruited 280 patients.[29] Although results are not yet published in full, the FOCUS trial met its primary endpoint with patients in the FOS arm having a reduced number of days on supplemental oxygen compared with placebo (4.8 vs 7.6 days, respectively, p=0.0136). FOS also showed a trend in reducing COVID-19-related mortality and morbidity. Unlike MATIS, this study also recruited patients with more severe

COVID-19; all three participants with a baseline score of 6 on the 8-point COVID-19 Ordinal Scale in the placebo group died by day 30, and all three participants with this baseline score in the FOS group survived, suggesting there may be particular benefit for patients with more severe COVID-19, although numbers in this cohort remain very small.

The study designs were different for these trials, limiting comparisons, but overall, it is possible that the timing of the use of FOS in diseases such as COVID-19 is critical, with greater benefit in patients with more severe disease, preventing the progression of lung disease, rather than earlier in the infection, when antiviral responses may be more important.

## Strength and limitations
### Patient population and recruitment
Despite a strong study design, we were unable to recruit the number of participants needed to detect a treatment effect for RUX or FOS due to reducing numbers of patients being admitted to the hospital as the pandemic evolved. Recruitment to MATIS was also limited by competition from other large concurrent COVID-19 trials. Another limitation is the representativeness of patients in the trial in terms of the age of the patients recruited. Older patients, who had the highest mortality in COVID-19, although approached for MATIS, were less likely to consent to the trial. We do not know the reason for this, but possible causes include family members declining study; fear of the medical profession; fear of the virus; patients being too unwell to read the patient information sheet; flaws in recruitment strategies in a fast-moving time. Recruitment of participants was predominantly from one large teaching hospital in London with a small number recruited from hospitals in other UK cities. Patients were admitted to this hospital through general acute medicine pathways rather than through any tertiary referral mechanism. We therefore consider the study population to be broadly representative of the general urban UK population requiring hospital admission for treatment of COVID-19. The demographics of participants showed broad ethnic representation (approximately 40% of participants were white and 60% were non-white), reflecting the recruitment from inner city hospitals in the UK and was a strength of the study.

The target population in MATIS was restricted to hospitalised patients. This included patients admitted with escalating symptoms and/or with a higher risk of progression to more severe disease due to underlying comorbidities. The trial was not designed to explore whether these treatments may offer any benefits to the wider community of patients with milder COVID-19 or to those who did not seek or receive hospital care.

### Evolving SOC
At or prior to randomisation, 46.0–48.4% of patients in MATIS received antiviral agents, 88.9–96.6% received immune-modulating agents, 58.6–74.6% received

antibiotics, 84.4–93.1% received anticoagulation treatment and 1.6–10.3% received antiplatelet therapy. The frequency of use of these agents was similar between trial arms, but we did not examine whether there were any differences in the numbers of agents or doses used. However, the SOC for treating COVID-19 changed over time, reflecting the dynamic nature of the evolving pandemic. Initially, treatments were focused on supportive care strategies. As the SOC evolved, antiviral medications, anti-inflammatory agents and eventually vaccination campaigns were integrated into treatment protocols. In 2021, there was a successful roll-out of vaccines. The changes in recommended treatments over distinct phases of the pandemic created additional complexity to COVID-19 treatment clinical trials impacting on patient profile, recruitment numbers and the changing use of multiple immunomodulatory agents. Within MATIS, the use of IL-6 inhibitors was introduced at Imperial College NHS Trust (the main study site) for patients who were developing hyper-inflammation. We note that fewer patients in the active treatment arms required treatment with an IL-6 inhibitor post randomisation.

### Implications for practice and future research

One of the unique contributing factors to the high individual mortality, as well as the population-level impact of COVID-19, was the triggering of a hyperinflammatory state causing a rapid escalation to ventilation in many patients. The excessive need for ventilation also impacted the rest of the population with hospitals running out of ventilators and intensive care staff.

In designing MATIS, we chose two different immune modulators as early interventions (within the first few days of admission), aiming to reduce inflammation, prevent the requirement for ventilation and improve the outcome for these patients, as well as reducing the impact of COVID-19 on UK hospitals.

To identify patients who we thought were at risk of hyperinflammation, and therefore more likely to benefit from immune modulation, we included patients with a raised CRP. However, this may not be specific enough. The RuxCoFlam trial, which showed a significant effect of RUX, selected patients based on a composite COVID-19 Inflammation Score (CIS) made up of multiple inflammatory markers.[30] This included inter alia CRP, ferritin, triglycerides, fibrinogen, total white blood cell count, lymphocytes and D-dimer and activated partial thromboplastin time. For CRP to contribute to the CIS, values were required to exceed 20 times the upper limit of the normal reference range. Subsequent research, including evidence from our own studies, has shown the importance of host genetic factors in COVID-19 mortality, demonstrating an exaggerated hyperinflammatory response to this novel virus.[31 32] Testing patients for these host factors, and measuring specific immune responses, may allow us to target patients more effectively for immune modulation and clarify which modulators are most appropriate at an individual level. A biomarker study is now underway

in samples stored prior to randomisation of MATIS to see if we can better fine-tune immune modulation therapy in virally induced hyperinflammation. Other patient characteristics such as age, obesity, diabetes and cardiovascular risk factors are independently associated with the development of secondary infection thrombosis, requirement for ventilation and death.[33] Meta-analysis of JAK/STAT inhibition from all studies may help identify the cohorts of patients most likely to benefit from these treatments.

The timing and choice of immune modulation in diseases such as COVID-19 could be critical for improving outcome, giving the immune system time to effectively eradicate the virus, but preventing the rapid escalation to hyperinflammation.

The results of these studies are likely to be important in guiding the management of other virally induced multisystemic disorders.

## CONCLUSION

In conclusion, we did not observe a significant treatment effect for FOS in comparison to SOC in reducing the proportion of hospitalised patients progressing to severe COVID-19 pneumonia. Results for RUX could not be confirmed due to early termination of our trial. Other factors such as patient comorbidities, choice of dose and timing of treatment may also have contributed to these findings. Further research is needed to help us to further understand these results and further investigation of the role of JAK/STAT inhibition in hyperinflammatory conditions triggered by viruses is now warranted, as well as long-term follow-up of patients who received immune modulation.

**Author affiliations**
[1]Imperial College Clinical Trials Unit, Imperial College London, London, UK
[2]Department of Immunology and Inflammation, Imperial College London, London, UK
[3]Imperial College Healthcare NHS Trust, London, UK
[4]Population Health Sciences Institute, Newcastle University, Newcastle upon Tyne, UK
[5]Immunology, Leeds Teaching Hospitals NHS Trust, Leeds, UK
[6]University of Leeds, Leeds, UK
[7]Infectious Diseases and Acute Medicine, Northwick Park Hospital, Harrow, UK
[8]Haemato-Oncology, Royal Berkshire NHS Foundation Trust, Reading, UK
[9]Department of Infectious Diseases, Sheffield Teaching Hospitals NHS Foundation Trust, Sheffield, UK
[10]Respiratory Medicine, Imperial College Healthcare NHS Trust, London, UK
[11]National Heart and Lung Institute, Imperial College London, London, UK
[12]Department of Metabolism, Digestion and Reproduction, Imperial College London, London, UK
[13]Department of Infectious Diseases, Imperial College London, London, UK

**Acknowledgements** The authors would like to thank the patients (and their families) who generously gave their time to take part in this study. We acknowledge the invaluable contributions of all of the research staff and coordinators at all participating sites, with special mention to Sophie Ryder, Clive Matthews, Simran Johal and Zayneb Alsaadi (Hammersmith Hospital, London); Ellen Calvelo, and Emma Thorley (St Mary's Hospital, London); Sanjay Mistry and Emily White (Charing Cross Hospital, London). We would like to acknowledge the contributions of the nurses and clinicians who assisted with patient recruitment at participating hospitals including but not limted to Anna Daunt, Adam Daneshmend, Tamara Elliot, Chloe Harding, Medha Kanitkar, Alexandra Mundell, Thomas Newman, Harsimranjit

Sidhu, Elizabeth Taylor, Emma Tilney, Eleanor Watson and Louisa White. Special thanks for laboratory oversight to Margarita Dominguez-Villar; for pharmacy support to Farah Abdel-Aziz; for administrative support to Camelia Vladescu, and Maria Martinez; and Sapna Dave for statistical support in preparing the study results. We would like to thank the members of the trial steering committee, the data management committee and the trial management group for their oversight and guidance throughout the trial.

**Contributors** NC conceived the research idea and led the trial. NC attained funding for the trial. NC, CP, SS, AC, AW, PN, PC, LC, MW, DM, OMK, TY, AI, MT, GSC and NV were principal investigators or investigators and contributed to the inception, trial management, data collection and/or completion of the trial. RP, JW, VC and NC were responsible for the study design and statistical analysis plan. LH, SC and RP conducted the statistical analysis. LH wrote the first draft of the manuscript with input from RP, VC, CP and NC. All authors approved the submitted version. NC, as guarantor, accepts full responsibility for the work and/or the conduct of the study, had access to the data and controlled the decision to publish.

**Funding** The Multi-Arm Trial of Inflammatory Signal Inhibitors for COVID-19 trial was sponsored by Imperial College London and was jointly funded by National Institute for Health and Care Research Imperial Biomedical Research Centre, Novartis, the manufacturer of ruxolitinib (Jakavi) and Rigel, the manufacturer of fostamatinib (Tavlesse). The sponsor and funders had no control over the study design; data collection, analysis or interpretation of data; in the writing of the report; or in the decision to submit the article for publication. All authors confirm independence from the funders, had full access to all of the data (including statistical reports and tables) and take responsibility for the integrity of the data and the accuracy of the data analysis. The open access fee was paid from the Imperial College London Open Access Fund.

**Competing interests** All authors have completed the Unified Competing Interest form (available on request from the corresponding author). LH, CP, RP, AW, PN, PC, LC, MT and NV declare no support from any organisation for the submitted work; no financial relationships with any organisations that might have an interest in the submitted work in the previous 3 years and no other relationships or activities that could appear to have influenced the submitted work. VC is a coinvestigator leading statistical design, methods and analysis for the MATIS trial, funded by Novartis and Rigel, is a committee member for the National Institute for Health and Care Research (NIHR) Academy Doctoral board, a DMEC member for the NIHR HTA-funded BAY Project. Behavioural Activation for young people with depression in specialist child and adolescent mental health services and is chair of the DMC for NIHR HTA-funded Research into Antipsychotic Discontinuation and Reductions. AC has been an employee of Roche since 2022. DM has received honorary payments from Novartis, Incyte and Ascentage Pharma, has participated on an advisory board for Ascentage Pharma and has received investigator-led institutional research funding from Incyte. MW has received honorary payments from AstraZeneca, has participated on an advisory board or data monitoring committee for Hansa Pharmaceuticals, received equipment, materials, drugs, medical writing, gifts or other services from Moderna and has had a leadership or fiduciary role in other board, society, committee or advocacy group, paid or unpaid for Hansa Pharmaceuticals. OMK has participated on an advisory board or data monitoring committee for the NIHR DILI study and has had team members funded by grants from Cepheid. GSC has participated in an end point review committee for Sanofi and is a board member for the MHRA. SS holds the Chair of British Society for Immunology-Clinical Immunology professional network. JW and SC have had team members funded by grants from Novartis and Rigel. TY is the recipient of a grant from UKRI MRC BHF Cardio-IMID Partnership, has received consulting fees for the Ad Board Abbvie 2024 and is Treasurer for the UK and Ireland Vasculitis and Rare Disease Group. AI has received consulting fees, honouraria for speakers' bureaus, international travel support and/or personal fees from Novartis and Incyte. NC has received research grants from Novartis and Rigel for conducting MATIS, honouraria for consultancy from Novartis, Sobi, Sanofi, Amgen, Argenyx, research support from Novartis and Argenx and has funding from the NIHR Imperial Biomedical Research Centre.

**Patient and public involvement** Patients and/or the public were not involved in the design, or conduct, or reporting, or dissemination plans of this research.

**Patient consent for publication** Not applicable.

**Ethics approval** This study involves human participants. Participants gave informed consent to participate in the study before taking part. The Surrey Research Ethics Committee (REC) and Health Regulator Authority (HRA) reviewed and granted approval for this trial (IRAS ID 282552). Informed consent was obtained for each participant either by the participant themselves or from a relative or an independent treating clinician acting as their legally designated personal representative in the event that the participant lacked capacity to provide consent.

**Provenance and peer review** Not commissioned; externally peer reviewed.

**Data availability statement** Data are available upon reasonable request. Deidentified participant-level data (excluding free text fields) and supporting documentation (including the MATIS study protocol, Statistical Analysis Plan and data dictionary) can be made available on reasonable request from the corresponding author (https://profiles.imperial.ac.uk/n.cooper) for the purposes of scientific research including secondary analysis of the data or for individual participant meta-analysis with appropriate human research ethics approvals and data transfer agreements in place.

**ORCID iDs**
Lorna Hazell https://orcid.org/0000-0002-5962-0648
James Wason https://orcid.org/0000-0002-4691-126X
Onn Min Kon https://orcid.org/0000-0003-2647-4688

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
