## [Reviewer comments · BMJ Open]

ARTICLE DETAILS

Title (Provisional)

Multi-arm multi-stage randomised controlled trial of Inflammatory Signal Inhibitors (MATIS) for patients hospitalised with COVID-19 pneumonia during the UK pandemic.

Authors

Hazell, Lorna; Pillay, Clio; Cornelius, Victoria; Phillips, Rachel; Charania, Asad; Wason, James; Cherlin, Svetlana; Savic, Sinisa; Whittington, Ashley; Neelakantan, Pratap; Collini, Paul; Cook, Lucy; Willicome, Michelle; Milojkovic, Dragana; Kon, Onn Min; Youngstein, Taryn; Innes, Andrew; Thursz, Mark; Cooke, Graham S; Vergis, Nikhil; Cooper, Nichola

VERSION 1 - REVIEW

Reviewer	1
Name	Siripongboonsitti, Taweegrit
Affiliation	Chulabhorn Royal Academy
Date	30-Apr-2025
COI	None

Comment 1:

Some typographical errors were identified throughout the manuscript. A careful proofreading is recommended.

Comment 2:

Could the authors provide baseline characteristics stratified by variant period (Alpha, Delta, and Omicron)? Existing data suggest that case fatality rate (CFR), disease severity, and hospitalization rates in unvaccinated individuals declined during the Omicron wave compared to the Alpha and Delta periods.

Comment 3:

Please include the time from symptom onset to treatment initiation, as this may influence disease progression and impact the study outcomes.

Comment 4:

Approximately half of the participants in each group received more than one antiviral agent, and most also received immunomodulatory therapy. These treatments may act as significant confounders and should be considered in the analysis of the primary outcomes.

Comment 5:

This study contributes valuable data on the treatment effect of favipiravir in non-severe COVID-19 patients classified as WHO clinical progression scale 3–4, an area that has not been well established in previous research.

Comment 6:

The authors should provide more discussion

Reviewer	2
Name	Leong, Christine
Affiliation	University of Manitoba
Date	24-May-2025
COI	None

This is an adaptive multi-stage open-label randomized trial conducted in 181 patients from 5 hospitals to evaluate the safety and efficacy of ruxolitinib and fostamatinib for 14 days compared to SOC for mild to moderate COVID-19 pneumonia. The primary outcome was severe COVID pneumonia within 14 days of randomization.

Trial Design and Participants, page 11. Could consider elaborating further on whether there is criteria for hospitalizing someone with only mild to moderate pneumonia. Were these patients at higher risk for progressing to more severe pneumonia (eg immunocompromised drug, uncontrolled copd/asthma, not vaccinated, older adults), or do all patients with COVID regardless of severity seek hospital care? I suspect there is a large proportion of patients who seek outpatient care not captured in this cohort which may influence results.

Intervention, page 13. Comment on whether all interventions were given by mouth (specify route of admin). Were there patients unable to take intervention due to not being able to take meds by mouth. Did patient receive all doses.

Table 2. Could report on baseline inflammatory markers

Table 3. Drug exposure - could describe time period of exposure to at least one dose (during randomization period vs prior to hospitalization). Also could comment on doses (high vs low) and if not available could note in limitations.

Reviewer	3
-----------------	----------

Name Karibayeva, Indira
Affiliation Georgia Southern University, Department of Health Policy and Community Health
Date 15-Jul-2025
COI None

The article is fundamentally strong and addresses an important clinical and methodological issue

Reviewer 4
Name Kelson, Zoe
Affiliation University of Exeter, Mathematics
Date 19-Aug-2025
COI None

This study aims to determine the safety and efficacy of ruxolitinib and fostamatinib compared to standard of care (SOC) for the treatment of mild to moderate COVID-19 pneumonia.

Reviewer comments:

"The MATIS trial was registered prospectively in ClinicalTrials.gov (NCT04581954 and EUDRACT (2020-001750-22) before enrolling the first participant."

The trial has been suitably registered.

Can the authors please provide a copy of the pre-specified study protocol and statistical analysis plan (SAP)?

"Participant flow through the trial was described using a Consolidated Standards of Reporting Trials (CONSORT) flow chart."

Thanks for providing a copy of the CONSORT checklist.

"Adaptive multi-arm, multi-stage, randomised, open label trial (3-arm, 2- stage)" [Abstract] and

"The Multi-Arm Trial of Inflammatory Signal Inhibitors for COVID-19 (MATIS) was an open-label multi-arm, multi-stage randomised (1:1:1) controlled trial (RCT) of ruxolitinib (RUX) and fostamatinib (FOS) for COVID-19 pneumonia compared to routine standard of care (SOC)"

A rigorous study design was developed.

"Five hospitals in England between October 2020 and September 2022"

Can the authors please comment on whether these five hospitals can be considered to be representative?

"At Stage 1, 181 patients were randomised, with 4 assessed as ineligible post-randomisation. Fostamatinib was stopped early for futility with 16 participants (27.6%, N=58) developing severe COVID pneumonia compared to 15 (25.0%, N=60) in the SOC arm (adjusted odds ratio (aOR) compared to SOC: 1.12; 95% confidence interval (CI): 0.49 to 2.58; p value = 0.608). Ruxolitinib progressed to Stage 2 but the trial was stopped early due to slow recruitment. "

and

"A planned formal interim analysis (Stage 1) was conducted on 10th March 2022 after 177 eligible participants had been randomised and completed 14 days of follow-up. A prespecified threshold of one-sided p-value ≤ 0.25 was required to progress treatment with RUX and/or FOS to Stage 2. The Independent Data Monitoring Committee (IDMC) recommended stopping the FOS arm for futility at Stage 1 (see Results section). Recruitment continued for the RUX and SOC arms, however, due to the slow recruitment because of the reduction in new COVID-19 cases, the trial was stopped early on 2nd September 2022 after only recruiting an additional 4 participants. Therefore, the final analysis presented in this manuscript includes all three trial arms as if they had been a conventional one-stage RCT comparing RUX and FOS with SOC. "

The authors transparently state the limitations and early termination of the trial.

"A sample size of 171 (57 per arm) participants at Stage 1 and an additional maximum of 285 (95 per arm) was selected to provide power of 90% (minimum marginal power) with a maximum 5% chance of an intervention arm being recommended when it provides no improvement over control (5% one-sided familywise error rate), inflated by 5% for missing outcome data (full details are described in the trial protocol)"

Can effect sizes please be reported for this sample size calculation and assessment of study power?

"For the primary outcome, we used a generalized linear model with a binomial distribution and logit link function to compare the odds of developing severe COVID pneumonia between RUX or FOS and SOC. At Stage 1, two separate models were used to compare the effects of FOS vs SOC and RUX vs SOC. These models were adjusted for using a propensity score derived from a logistic regression model that included baseline WHO COVID-19 severity grade (3 or 4), age (<65 or ≥ 65), receipt of steroids at baseline and receipt of interleukin-6 (IL6) inhibitors at baseline. For the final analysis, we included all three trial arms in a single model. Baseline WHO COVID-19 severity grade (3 or 4), age (<65 or ≥ 65), receipt of interleukin-6 (IL6) inhibitors at the time of or prior to randomisation and prior vaccination against COVID-19 were included as separate covariates in this model. As there were several sites with very few randomised participants, we excluded study site from the primary analysis models"

and

"COVID severity grade was modelled as an ordinal variable using a mixed ordinal logistic regression model to compare the odds of progression to more severe disease at Day 14. Mixed linear regression models were used to compare changes in inflammatory markers and serum creatinine up to Day 28 and a Cox proportional hazards model was used to compare the rates of hospital discharge between trial arms. The frequency of venous thromboembolism, readmission within 28 days and SAE were summarised descriptively. Dot plots were used to visually display between group differences in SAE frequencies (as risk ratios) grouping events at system organ class level."

Appropriate modelling methods have been applied.

"Participants were randomly allocated to RUX, FOS or SOC using a central webbased randomisation service that used randomisation sequences with random block sizes stratified by age (<65 vs ≥65) and study site."

Given site is a stratification variable for the randomisation, can this please be adjusted for in the models?

Furthermore, can models please also adjust for Sex?

"Multiple imputation was used to account for any missing data such that all eligible randomised participants were included in both the Stage 1 and final analyses (See Supplementary Appendix 3)"

The authors suitably assess missing data, and undertake sensitivity analyses on their approach to handling it.

"However, as the planned sample size was not reached, the p-values presented should not be interpreted with reference to this threshold as the trial was significantly underpowered. Interpretation should be focused on the point estimate and 95% confidence interval as a way to assess the evidence of a treatment effect. "

Can the authors please clarify that a treatment effect cannot be inferred from the analysis due to the study being underpowered? Given the early termination of the trial and subsequent small sample sizes, analyses should be reported and inferred as descriptive and exploratory only.

"Sensitivity analyses were undertaken to examine alternative approaches to handling missing data firstly, assuming that all participants with missing outcome had poor outcome e.g., all missing assumed severity grade ≥5, and secondly assuming that all participants with missing outcome had good outcome e.g., all missing assumed severity grade <5. Additional exploratory analysis assessing the impact of participant adherence¹², time of enrolment to the trial and subgroup analysis of the primary outcome by age, sex, BMI, cardiovascular disease, chronic lung disease, end-stage renal failure, liver cirrhosis, diabetes, baseline CRP

and D-dimer, immunocompromisation, smoking status and number of days since onset of COVID at randomisation are briefly described in the Supplementary Appendices 5 and 6)."

A number of insightful additional analyses have been explored.

"Results from the per-protocol analysis in participants who received $\geq 90\%$ of prescribed dose were consistent with the primary analysis (Supplementary Table S13)."

Can this sensitivity analysis please be specified in the methods section?

"At the final analysis, a lower proportion of participants developed severe COVID pneumonia in the ruxolitinib arm (n=10, 16.1%, N=62) than in the SOC arm (15, 24.6%, N=61) (aOR: 0.63; 95% CI: 0.25 to 1.57; p=0.161)"

and

"The ruxolitinib treatment effect estimate was in favour of benefit, but due to early stopping, the trial was underpowered to confirm ruxolitinib's benefit in this population"

and

"Despite not having power for a definitive outcome, there was a suggestive finding indicating lower odds (OR: 0.63) of progressing to severe COVID-19 pneumonia (grade ≥ 5) associated with ruxolitinib compared to SOC, even though the wide 95% confidence interval (0.25 to 1.57) and a p-value of 0.161 warrant cautious interpretation of the findings. In contrast, there was no suggestive finding indicating lower odds for fostamatinib. In fact, the odds were higher (OR 1.19) for progressing to severe COVID-19 pneumonia associated with fostamatinib compared to SOC. Although, again, there was a wide 95% confidence interval (0.51 to 2.76) and a pvalue of 0.659"

Can the authors please remove inferences for these findings (i.e. 'lower' and 'in favour') given the CIs cross 1 and the outcomes are not statistically significant?

"Despite a strong study design, we were unable to recruit the numbers of participants needed to detect a treatment effect for ruxolitinib or fostamatinib due to reducing numbers of patients being admitted to hospital as the pandemic evolved. Recruitment to MATIS was also limited by competition from other large concurrent COVID trials. Another limitation is the representativeness of patients in the trial in terms of age of the patients recruited. Older patients, who had the highest mortality in COVID-19, although approached for MATIS, were less likely to consent to the trial. We do not know the reason for this, but possible causes include family members declining study; fear of the medical profession; fear of the virus; patients being too unwell to read the patient information sheet; flaws in recruitment strategies in a fastmoving time. On the other hand, the demographics of participants showed broad ethnic representation (approximately 40% of participants were white and 60% were non-white), reflecting the recruitment from inner city hospitals and was a strength of the study"

A discussion on the strengths and limitations of the study has been provided by the authors.

VERSION 1 - AUTHOR RESPONSE

Reviewer 1

Dr. Taweegrit Siripongboonsitti, Chulabhorn Royal Academy

Comments to the Author:

Comment 1

Some typographical errors were identified throughout the manuscript. A careful proofreading is recommended.

Author response:

We have proof-read and corrected any errors noted.

Comment 2

Could the authors provide baseline characteristics stratified by variant period (Alpha, Delta, and Omicron)? Existing data suggest that case fatality rate (CFR), disease severity, and hospitalization rates in unvaccinated individuals declined during the Omicron wave compared to the Alpha and Delta periods.

Author response:

Thank you for this question. The MATIS trial recruitment period spanned October 2020 to September 2022. This mainly covered the Alpha and Delta waves. Although Omicron was detected from around December 2021 in the UK, only 12 participants were recruited to MATIS after that date. Indeed, as the Omicron variant was associated with less severe disease, this (along with the impact of the vaccine roll-out) contributed to the recruitment challenges for the trial as fewer people were being hospitalised.

Below we have provided a table of baseline characteristics of participants stratified by the predominant variant at the time of enrolment. Our observations here are as expected, e.g., the % of patients with severity Grade 4 reduced across the different variants and the % vaccinated increased. Both of these baseline characteristics were adjusted for in our analysis model. We have added this table to the Appendices and added some text to reflect our observations in the Results section:

*“Recruitment followed the peaks and troughs of the COVID-19 pandemic waves observed in the UK during the trial period (Figure 3). **This period predominantly covered the Alpha and Delta waves in the UK. Although the Omicron variant was detected from around December 2021, only 12 participants were recruited to MATIS after that date.**”*

*“**The percentage of participants with a baseline COVID-19 severity grade of 4 decreased over the trial recruitment period from 84.8% in the Alpha wave to 75% for those recruited in the Omicron wave, while the proportion of participants who had received the COVID-19 vaccination increased from 8% to 50% (Table S1).**”*

To explore differences in treatment effects by calendar time we performed a pre-specified exploratory analysis of the primary outcome stratified by quartile of

randomisation dates. As shown in Supplementary Appendix (Table S14) and described on p28 of the manuscript, the numbers in each group were too small to show any difference between randomisation periods with imprecise effect estimates within each stratum.

Table S1: Baseline characteristics of all participants by predominant variant at time of randomisation:

Patient characteristic, n(%) unless otherwise specified	Alpha (up to May 2021) (N=112)	Delta (May to Dec 2021) (N=61)	Omicron (post Dec 2021) (N=12)
Age in years			
Mean (SD)	60.8 (14.7)	55.9 (16.6)	65.8 (17.1)
Sex			
Male	79 (70.5)	43 (70.5)	6 (50.0)
Female	33 (29.5)	18 (29.5)	6 (50.0)
Ethnicity			
White	41 (36.6)	30 (49.2)	8 (66.7)
Mixed or multiple ethnic groups	1 (0.9)	1 (1.6)	0 (0.0)
Asian or Asian British	14 (12.5)	8 (13.1)	4 (33.3)
Black, Black British, Caribbean or African	12 (10.7)	7 (11.5)	0 (0.0)
Other ethnic group	44 (39.3)	15 (24.6)	0 (0.0)
BMI in kg/m ²			
N (N missing)	106 (6)	55 (6)	11 (1)
Mean (SD)	30.2 (6.4)	31.0 (7.8)	26.5 (5.1)
Severity of Covid			
Grade 3	17 (15.2)	13 (21.3)	3 (25.0)
Grade 4	95 (84.8)	48 (78.7)	9 (75.0)
Chronic lung disease			
N (N missing)	111 (1)	60 (1)	12 (0)
Yes	15 (13.5)	13 (21.7)	2 (16.7)
Time from onset of symptoms			
N (N missing)	103 (9)	51 (10)	11 (1)
Mean (SD)	9 (4)	10 (4)	12 (5)
Diabetes			
N (N missing)	111 (1)	60 (1)	12 (0)
Yes	37 (33.3)	13 (21.7)	4 (33.3)
Hypertension			
N (N missing)	111 (1)	60 (1)	12 (0)
Yes	53 (47.7)	21 (35.0)	7 (58.3)
Ischaemic heart disease			
N (N missing)	111 (1)	60 (1)	12 (0)
Yes	17 (15.3)	9 (15.0)	5 (41.7)
Heart failure			
N (N missing)	111 (1)	60 (1)	12 (0)
Yes	3 (2.7)	2 (3.3)	2 (16.7)
Immunocompromised			
N (N missing)	111 (1)	60 (1)	12 (0)
Yes	4 (3.6)	2 (3.3)	2 (16.7)
End-stage renal failure			

N (N missing)	111 (1)	60 (1)	12 (0)
Yes	10 (9.0)	2 (3.3)	1 (8.3)
Liver cirrhosis			
N (N missing)	111 (1)	60 (1)	12 (0)
Yes	0 (0.0)	0 (0.0)	0 (0.0)
Current smoker			
N (N missing)	110 (2)	60 (1)	12 (0)
Yes	5 (4.5)	0 (0.0)	0 (0.0)
Prior Covid Vaccination			
N (N missing)	112 (0)	61 (0)	12 (0)
Yes	9 (8.0)	20 (32.8)	6 (50.0)
Serum creatinine ($\mu\text{mol/L}$)			
N (N missing)	111 (1)	59 (2)	12 (0)
Mean (SD)	139 (209)	95 (85)	115 (112)
C-reactive protein (mg/L)			
N (N missing)	111 (1)	59 (2)	12 (0)
Mean (SD)	115 (70)	100 (75)	76 (46)
Lactate dehydrogenase (IU/L)			
N (N missing)	73 (39)	28 (33)	8 (4)
Mean (SD)	478 (255)	388 (153)	390 (173)
Ferritin ($\mu\text{g/L}$)			
N (N missing)	98 (14)	43 (18)	10 (2)
Mean (SD)	1659 (2343)	1235 (1145)	472 (285)
D-dimer (ng/ml)			
N (N missing)	103 (9)	49 (12)	9 (3)
Mean (SD)	1851 (3302)	996 (764)	1391 (1082)

Comment 3

Please include the time from symptom onset to treatment initiation, as this may influence disease progression and impact the study outcomes.

Author response:

Thank you for this suggestion. This was similar in all three trials arms (around 10 days). We have added this information for context to Table 2.

Comment 4

Approximately half of the participants in each group received more than one antiviral agent, and most also received immunomodulatory therapy. These treatments may act as significant confounders and should be considered in the analysis of the primary outcomes.

Author response:

To clarify, the interpretation from Table 2 is that just under half of participants in all three trial arms received one or more (≥ 1) dose of antiviral agent prior to randomisation. The antiviral received was almost exclusively remdesivir which did not have proven efficacy at the time of the trial.

We have added the following text to p16 of the manuscript to further explain the choice of covariates for inclusion in our analysis model:

“We planned to adjust our final analysis model for site, age category (<65 or ≥65), baseline COVID-19 severity grade (3 or 4) and receipt of treatments assumed to be effective prior to randomisation based on current knowledge at that time of the MATIS trial. These effective treatments included dexamethasone, IL-6 inhibitors, selected antivirals and COVID vaccination. As the efficacy of remdesivir was unconfirmed at the time of the trial, Paxlovid® and/or molnupiravir were the only antivirals specified a-priori, however, no participants received these agents. As a large proportion of participants had received dexamethasone in each arm, prior steroid use was not included in our final analysis model to avoid instability due to zero cell counts in some strata. Similarly, as there were several sites with very few randomised participants, we excluded study site from the primary analysis model. Thus, our final model adjusted for prior use of IL-6 inhibitors, age category, COVID-19 severity grade and receipt of COVID-19 vaccine. These were included as separate covariates in the final model.”

Comment 5

This study contributes valuable data on the treatment effect of favipiravir in non-severe COVID-19 patients classified as WHO clinical progression scale 3–4, an area that has not been well established in previous research.

Author response:

We thank the reviewer for this acknowledgement (we assume the reviewer has made a typo in their comment as “favipiravir” was not studied in MATIS).

Comment 6

The authors should provide more discussion

Author response:

We believe we have covered the important elements in our Discussion section including a summary of the main findings, the context of our results with other studies and mechanistic work, the strength and limitations and the potential implications of the results including the need for further research. In response to other reviewer comments, additional text has been added regarding treatment adherence (p29-30), the patient population (p33-34) and study limitations (p34).

Reviewer: 2

Dr. Christine Leong, University of Manitoba

Comments to the Author:

Comment 1

This is an adaptive multi-stage open-label randomized trial conducted in 181 patients from 5 hospitals to evaluate the safety and efficacy of ruxolitinib and fostamatinib for 14 days compared to SOC for mild to moderate COVID-19 pneumonia. The primary outcome was severe COVID pneumonia within 14 days of randomization.

Trial Design and Participants, page 11. Could consider elaborating further on whether there is criteria for hospitalizing someone with only mild to moderate pneumonia. Were these patients at higher risk for progressing to more severe pneumonia (eg immunocompromised drug, uncontrolled copd/asthma, not vaccinated, older adults), or do all patients with COVID regardless of severity seek hospital care? I suspect there is a large proportion of patients who seek outpatient care not captured in this cohort which may influence results.

Author response:

Thank you for this question. We have made minor edits to the 'Introduction' and 'Trial design and participants' sections to clarify that the trial population included only hospitalised patients with COVID severity Grade 3 or 4 on the modified World Health Organisation (WHO) severity scale, rather than any patient with severity Grade less than 5. For clarity we have removed the terminology of 'mild to moderate' COVID-19 pneumonia throughout the manuscript to avoid confusion. We are not aware of specific criteria for admission to hospital but acknowledge that some sicker patients may not have sought or received hospital care.

The following text has also been added to our limitations section in the Discussion

"The target population in MATIS was restricted to hospitalised patients. This included patients admitted with escalating symptoms and/or with a higher risk of progression to more severe disease due to underlying comorbidities. The study was not designed to explore whether these treatments may offer any benefits to the wider community of patients with milder COVID-19 disease or to those that did not seek or receive hospital care."

Comment 2

Intervention, page 13. Comment on whether all interventions were given by mouth (specify route of admin). Were there patients unable to take intervention due to not being able to take meds by mouth. Did patient receive all doses.

Author response:

Thank you for this suggestion. We did not systematically capture route of administration for each patient, however, we have added the following text to the Intervention section of the Methods.

"Treatment was given by mouth unless the participant was unable to take oral medicines, in which case treatments were administered via a nasogastric tube."

In the Methods section (page 15) we state "Participants were assumed to have received 14 days of treatment unless there was evidence for stopping earlier" and in the Results section (page 19) we state *"Thirty-eight (66.7%) participants in the FOS arm and 46 (71.9%) in the RUX arm were assumed to have completed the 14-day treatment course"*.

We were unable to reliably capture whether patients received all doses. Participants were advised to continue their prescribed medication on discharge from hospital but, due to the pandemic setting and social distancing protocols, participants were not expected to return for study follow up in-person. It was therefore not possible to reliably measure compliance or adherence as this was based on self-report rather than more reliable measures such as pill counts. We have added the following text to the Discussion section (p28/9) to acknowledge this limitation in the context of our results.

“Approximately one third of participants did not complete the 14-day treatment course for fostamatinib and ruxolitinib. The results of our complier-adjusted case analyses, which estimated the impact of adherence on the treatment effects, were consistent with our primary analysis. However, due to the pandemic setting, adherence measures were based on self-report for participants given take-home medication following hospital discharge rather than in-person pill counts. Our estimate of adherence and its impact on treatment effects may therefore be under-estimated.”

Comment 3

Table 2. Could report on baseline inflammatory markers

Author response:

Thank you for this suggestion, we have updated Table 2 to summarise baseline CRP, LDH, ferritin and d-dimer levels by treatment arm.

Comment 4

Table 3. Drug exposure - could describe time period of exposure to at least one dose (during randomization period vs prior to hospitalization). Also could comment on doses (high vs low) and if not available could note in limitations.

Author response:

We have added the following text to the Results section to describe the exposure time detailed in Table 3.

“These standard of care treatments were initiated a median of one day before or on the same day as randomisation in all three trial arms and continued once randomised with similar durations in all three arms ranging from a median of 5 days for antivirals and 9 days for anticoagulants (Table 3).”

We did not prespecify or perform any analysis of concomitant medications by dose. We have added the following text to the Limitations section.

“The frequency and duration of use of these agents was similar between trial arms but we did not examine whether there were any differences in the numbers of agents or doses used.”

Reviewer 3

Dr. Indira Karibayeva, Georgia Southern University

Comments to the Author:

The article is fundamentally strong and addresses an important clinical and methodological issue.

Author response:

Thank you. No action.

Reviewer 4

Prof. Zoe Kelson, University of Exeter

Comments to the Author:

This study aims to determine the safety and efficacy of ruxolitinib and fostamatinib compared to standard of care (SOC) for the treatment of mild to moderate COVID-19 pneumonia.

Comment 1

"The MATIS trial was registered prospectively in ClinicalTrials.gov (NCT04581954 and EUDRA-CT (2020-001750-22) before enrolling the first participant."

The trial has been suitably registered.

Author response:

Thank you. No action

Comment 2

Can the authors please provide a copy of the pre-specified study protocol and statistical analysis plan (SAP)?

Author response:

The protocol has been published (this is cited on page 12 of the manuscript):(<https://trialsjournal.biomedcentral.com/articles/10.1186/s13063-021-05190-z>).

We have added the SAP as an additional Supplementary Appendix 1.

Comment 3

"Participant flow through the trial was described using a Consolidated Standards of Reporting Trials (CONSORT) flow chart."

Thanks for providing a copy of the CONSORT checklist.

Author response:

Thank you. No action.

Comment 4

"Adaptive multi-arm, multi-stage, randomised, open label trial (3-arm, 2- stage)" [Abstract] and "The Multi-Arm Trial of Inflammatory Signal Inhibitors for COVID-19 (MATIS) was an open-label multi-arm, multi-stage randomised (1:1:1) controlled trial (RCT) of ruxolitinib (RUX) and fostamatinib (FOS) for COVID-19 pneumonia compared to routine standard of care (SOC)" A rigorous study design was developed.

Author response:

Thank you. No action.

Comment 5

"Five hospitals in England between October 2020 and September 2022"

Can the authors please comment on whether these five hospitals can be considered to be representative?

Author response:

We have added the following text to the Results section (p19)

"The majority of participants were recruited from one London teaching hospital site (n=170) with a smaller proportion from other UK city hospitals (n=15)."

And to the Discussion (p32):

"Recruitment of participants was predominantly from one large teaching hospital in London with a small number recruited from hospitals in other UK cities. Patients were admitted to this hospital through general acute medicine pathways rather than through any tertiary referral mechanism. We therefore consider the study population to be broadly representative of the general urban UK population requiring hospital admission for treatment of COVID disease."

Comment 6

"At Stage 1, 181 patients were randomised, with 4 assessed as ineligible post-randomisation. Fostamatinib was stopped early for futility with 16 participants (27.6%, N=58) developing severe COVID pneumonia compared to 15 (25.0%, N=60) in the SOC arm (adjusted odds ratio (aOR) compared to SOC: 1.12; 95% confidence interval (CI): 0.49 to 2.58; p value = 0.608). Ruxolitinib progressed to Stage 2 but the trial was stopped early due to slow recruitment. "

and

"A planned formal interim analysis (Stage 1) was conducted on 10th March 2022 after 177 eligible participants had been randomised and completed 14 days of follow-up. A prespecified threshold of one-sided p-value ≤ 0.25 was required to progress treatment with RUX and/or FOS to Stage 2. *The Independent Data Monitoring Committee (IDMC) recommended stopping the FOS arm for futility at Stage 1 (see Results section). Recruitment continued for the RUX and SOC arms, however, due to the slow recruitment because of the reduction in new COVID-19 cases, the trial was stopped early on 2nd September 2022 after only recruiting an additional 4 participants. Therefore, the final analysis presented in this manuscript includes all three trial arms as if they had been a conventional one-stage RCT comparing RUX and FOS with SOC.* " The authors transparently state the limitations and early termination of the trial.

Author response:

Thank you. Note that we have moved the latter part of this paragraph in italics to the Results section.

Comment 7

"A sample size of 171 (57 per arm) participants at Stage 1 and an additional maximum of 285 (95 per arm) was selected to provide power of 90% (minimum marginal power) with a maximum 5%

chance of an intervention arm being recommended when it provides no improvement over control (5% one-sided familywise error rate), inflated by 5% for missing outcome data (full details are described in the trial protocol)"

Can effect sizes please be reported for this sample size calculation and assessment of study power?

Author response:

The following sentence has been added to the sample size section for clarification:

"This sample size calculation assumed a 50% rate of severe pneumonia in the standard of care arm and a reduction of this risk by an experimental arm to 30% (relative risk, RR, 0.6). "

Comment 8

"For the primary outcome, we used a generalized linear model with a binomial distribution and logit link function to compare the odds of developing severe COVID pneumonia between RUX or FOS and SOC. At Stage 1, two separate models were used to compare the effects of FOS vs SOC and RUX vs SOC. These models were adjusted for using a propensity score derived from a logistic regression model that included baseline WHO COVID-19 severity grade (3 or 4), age (<65 or ≥65), receipt of steroids at baseline and receipt of interleukin-6 (IL6) inhibitors at baseline. For the final analysis, we included all three trial arms in a single model. Baseline WHO COVID-19 severity grade (3 or 4), age (<65 or ≥65), receipt of interleukin-6 (IL6) inhibitors at the time of or prior to randomisation and prior vaccination against COVID-19 were included as separate covariates in this model. As there were several sites with very few randomised participants, we excluded study site from the primary analysis models"

and

"COVID severity grade was modelled as an ordinal variable using a mixed ordinal logistic regression model to compare the odds of progression to more severe disease at Day 14. Mixed linear regression models were used to compare changes in inflammatory markers and serum creatinine up to Day 28 and a Cox proportional hazards model was used to compare the rates of hospital discharge between trial arms. The frequency of venous thromboembolism, readmission within 28 days and SAE were summarised descriptively. Dot plots were used to visually display between group differences in SAE frequencies (as risk ratios) grouping events at system organ class level."

Appropriate modelling methods have been applied.

Author response:

Thank you. No action.

Comment 9

"Participants were randomly allocated to RUX, FOS or SOC using a central webbased randomisation service that used randomisation sequences with random block sizes stratified by age (<65 vs ≥65) and study site."

Given site is a stratification variable for the randomisation, can this please be adjusted for in the models?

Author response:

We have updated the text on p16 of the manuscript to explain that although we planned to adjust for site, there were several sites with very few randomised participants so we excluded study site from the primary analysis models.

Comment 10

Furthermore, can models please also adjust for Sex?

Author response:

We anticipated balance across all confounders due to the randomised design and, in line with best practice, planned to adjust only for design variables (as appropriate) and any prespecified variables that were considered to be strongly associated with the outcome to increase the efficiency of our treatment effect estimate. As sex was not pre-specified in our SAP as either a design variable or a variable expected to be strongly associated with the outcome, to remain in line with our prespecified SAP we would prefer not to additionally adjust for this.

(<https://trialsjournal.biomedcentral.com/articles/10.1186/1745-6215-15-139>)

Comment 11

"Multiple imputation was used to account for any missing data such that all eligible randomised participants were included in both the Stage 1 and final analyses (See Supplementary Appendix 3)"

The authors suitably assess missing data, and undertake sensitivity analyses on their approach to handling it.

Author response:

Thank you. No action.

Comment 12

"However, as the planned sample size was not reached, the p-values presented should not be interpreted with reference to this threshold as the trial was significantly underpowered. Interpretation should be focused on the point estimate and 95% confidence interval as a way to assess the evidence of a treatment effect. "

Can the authors please clarify that a treatment effect cannot be inferred from the analysis due to the study being underpowered? Given the early termination of the trial and subsequent small sample sizes, analyses should be reported and inferred as descriptive and exploratory only.

Author response:

Thank you for this suggestion. In our description of the primary outcome results for our final analysis we have mentioned in the text that p-values are nominal and also in the footnote to Table 4. We believe it remains valid to consider the point estimate and confidence interval to assess the evidence of a treatment effect as the lack of precision (through underpowering) is inherently captured here. We suggest that the term hypothesis-generating may be appropriate here and have edit the following text:

*"However, as the planned sample size was not reached, the p-values presented should not be interpreted with reference to this threshold as the trial was significantly underpowered. Interpretation should focus on the point estimate and 95% confidence interval as a way to assess the evidence of a treatment effect **and associated***

uncertainty, however, this interpretation should be regarded as hypothesis-generating only.”

Comment 13

"Sensitivity analyses were undertaken to examine alternative approaches to handling missing data firstly, assuming that all participants with missing outcome had poor outcome e.g., all missing assumed severity grade ≥ 5 , and secondly assuming that all participants with missing outcome had good outcome e.g., all missing assumed severity grade < 5 . Additional exploratory analysis assessing the impact of participant adherence¹², time of enrolment to the trial and subgroup analysis of the primary outcome by age, sex, BMI, cardiovascular disease, chronic lung disease, end-stage renal failure, liver cirrhosis, diabetes, baseline CRP and D-dimer, immunocompromisation, smoking status and number of days since onset of COVID at randomisation are briefly described in the Supplementary Appendices 5 and 6)."

A number of insightful additional analyses have been explored.

Author response:

Thank you. No action.

Comment 14

"Results from the per-protocol analysis in participants who received $\geq 90\%$ of prescribed dose were consistent with the primary analysis (Supplementary Table S13)."

Can this sensitivity analysis please be specified in the methods section?

Author response:

The methods of this analysis were originally described briefly as an exploratory analysis in the Supplementary Appendix. We have now moved this description of the methods from the Appendix to the main text.

Comment 15

"At the final analysis, a lower proportion of participants developed severe COVID pneumonia in the ruxolitinib arm (n=10, 16.1%, N=62) than in the SOC arm (15, 24.6%, N=61) (aOR: 0.63; 95% CI: 0.25 to 1.57; p=0.161)"

and

"The ruxolitinib treatment effect estimate was in favour of benefit, but due to early stopping, the trial was underpowered to confirm ruxolitinib's benefit in this population"

and

"Despite not having power for a definitive outcome, there was a suggestive finding indicating lower odds (OR: 0.63) of progressing to severe COVID-19 pneumonia (grade ≥ 5) associated with ruxolitinib compared to SOC, even though the wide 95% CI (0.25 to 1.57) and a p-value of 0.161 warrant cautious interpretation of the findings. In contrast, there was no suggestive finding indicating lower odds for fostamatinib. In fact, the odds were higher (OR 1.19) for progressing to severe COVID-19 pneumonia associated with fostamatinib compared to SOC. Although, again, there was a wide 95% CI (0.51 to 2.76) and a pvalue of 0.659"

Can the authors please remove inferences for these findings (i.e. 'lower' and 'in favour') given the CIs cross 1 and the outcomes are not statistically significant?

Author response:

Thank you for highlighting this. We have modified the language used and stated that the differences are numerical.

Abstract:

“At the final analysis, 10 participants (16.1%, N=62) developed severe COVID-19 pneumonia in the ruxolitinib arm compared to 15 (24.6%, N=61) in the SOC arm (aOR: 0.63; 95% CI: 0.25 to 1.57; p=0.161).”

“Progression to severe COVID-19 pneumonia was less frequent in the ruxolitinib arm, but due to early stopping, the trial was underpowered to confirm ruxolitinib’s benefit in this population.”

Discussion: "Despite being underpowered to make a definitive conclusion, a numerically lower odds (OR: 0.63) of progressing to severe COVID-19 pneumonia (grade ≥5) was observed with ruxolitinib compared to SOC. However, the 95% CI is wide (0.25 to 1.57) and crosses 1 indicating uncertainty and requires cautious interpretation. In contrast, the odds of progression were observed to be numerically higher (OR 1.19) with fostamatinib compared to SOC. Although, again, results were imprecise with a wide 95% confidence interval (0.51 to 2.76)."

Comment 16

"Despite a strong study design, we were unable to recruit the numbers of participants needed to detect a treatment effect for ruxolitinib or fostamatinib due to reducing numbers of patients being admitted to hospital as the pandemic evolved. Recruitment to MATIS was also limited by competition from other large concurrent COVID trials. Another limitation is the representativeness of patients in the trial in terms of age of the patients recruited. Older patients, who had the highest mortality in COVID-19, although approached for MATIS, were less likely to consent to the trial. We do not know the reason for this, but possible causes include family members declining study; fear of the medical profession; fear of the virus; patients being too unwell to read the patient information sheet; flaws in recruitment strategies in a fastmoving time. On the other hand, the demographics of participants showed broad ethnic representation (approximately 40% of participants were white and 60% were non-white), reflecting the recruitment from inner city hospitals and was a strength of the study"

A discussion on the strengths and limitations of the study has been provided by the authors.

Author response:

Thank you. No action

Reviewer: 1 competing interests.: Not applicable

Reviewer: 2 competing interests.: Not applicable

Reviewer: 3 competing interests.: Not applicable

Reviewer: 4 competing interests.: Not applicable

VERSION 2 - REVIEW

Reviewer **4**
Name **Kelson, Zoe**
Affiliation **University of Exeter, Mathematics**
Date **30-Oct-2025**
COI

Thanks to the authors for their responses and revisions.

Reviewer previous comment:

Can the authors please remove inferences for these findings (i.e. 'lower' and 'in favour') given the CIs cross 1 and the outcomes are not statistically significant?

Author response:

"Thank you for highlighting this. We have modified the language used and stated that the differences are numerical. Abstract: "At the final analysis, 10 participants (16.1%, N=62) developed severe COVID-19 pneumonia in the ruxolitinib arm compared to 15 (24.6%, N=61) in the SOC arm (aOR: 0.63; 95% CI: 0.25 to 1.57; p=0.161)." "Progression to severe COVID-19 pneumonia was less frequent in the ruxolitinib arm, but due to early stopping, the trial was underpowered to confirm ruxolitinib's benefit in this population."

Discussion: "Despite being underpowered to make a definitive conclusion, a numerically lower odds (OR: 0.63) of progressing to severe COVID-19 pneumonia (grade ≥ 5) was observed with ruxolitinib compared to SOC. However, the 95% CI is wide (0.25 to 1.57) and crosses 1 indicating uncertainty and requires cautious interpretation. In contrast, the odds of progression were observed to be numerically higher (OR 1.19) with fostamatinib compared to SOC. Although, again, results were imprecise with a wide 95% confidence interval (0.51 to 2.76)."

Reviewer comment:

Can the authors please revise the inferences and conclusions, noting that no association can be concluded here?

The use of 'numerical' language can be misleading, as the statistical analyses demonstrate no evidence to conclude a difference. Additionally, language such as 'less frequent' and 'lower odds' should be removed when the OR CIs cross 1. Furthermore, noting that the study is underpowered should not imply that a statistical difference may/may not exist if suitably powered, but rather that outcomes are subject to Types I and II error and should therefore be reported as descriptive only (i.e. without statistical inference).

VERSION 2 - AUTHOR RESPONSE

Reviewer #4 previous comment:

Can the authors please remove inferences for these findings (i.e. 'lower' and 'in favour') given the CIs cross 1 and the outcomes are not statistically significant?

Author response:

"Thank you for highlighting this. We have modified the language used and stated that the differences are numerical. Abstract: "At the final analysis, 10 participants (16.1%, N=62) developed severe COVID-19 pneumonia in the ruxolitinib arm compared to 15 (24.6%, N=61) in the SOC arm (aOR: 0.63; 95% CI: 0.25 to 1.57; p=0.161)." "Progression to severe COVID-19 pneumonia was less frequent in the ruxolitinib arm, but due to early stopping, the trial was underpowered to confirm ruxolitinib's benefit in this population."

Discussion: "Despite being underpowered to make a definitive conclusion, a numerically lower odds (OR: 0.63) of progressing to severe COVID-19 pneumonia (grade ≥ 5) was observed with ruxolitinib compared to SOC. However, the 95% CI is wide (0.25 to 1.57) and crosses 1 indicating uncertainty and requires cautious interpretation. In contrast, the odds of progression were observed to be numerically higher (OR 1.19) with fostamatinib compared to SOC.

Although, again, results were imprecise with a wide 95% confidence interval (0.51 to 2.76)."

Reviewer #4 comment:

Can the authors please revise the inferences and conclusions, noting that no association can be concluded here?

The use of 'numerical' language can be misleading, as the statistical analyses demonstrate no evidence to conclude a difference. Additionally, language such as 'less frequent' and 'lower odds' should be removed when the OR CIs cross 1. Furthermore, noting that the study is underpowered should not imply that a statistical difference may/may not exist if suitably powered, but rather that outcomes are subject to Types I and II error and should therefore be reported as descriptive only (i.e. without statistical inference).

Author response:

Thank you. We have made the following edits to the following sections. We trust that these changes are acceptable.

Abstract - Conclusion:

"We found no evidence that fostamatinib was superior to SOC for the treatment of COVID-19 pneumonia in patients requiring hospital admission. ~~Progression to severe COVID-19 pneumonia was less frequent in the ruxolitinib arm, but~~ Due to early stopping, the trial was underpowered to ~~confirm~~ establish ruxolitinib's ~~benefit~~ effect in this population. Further study is needed."

Discussion:

MATIS was an open-label, multi-centre, multi-arm, and multi-stage randomised controlled trial of ruxolitinib and fostamatinib for COVID-19 pneumonia compared to routine standard of care. The study was designed between March and April 2020, with

participants recruited largely from the second and third waves of the UK pandemic, with the majority of participants recruited from three hospitals in North-West London. Our efficient, multi-stage, multi-arm design allowed early stopping of the fostamatinib arm for futility. This design allows recruitment efforts to focus on more promising treatments, a particularly important feature in the pandemic setting. However, due to the success of the COVID-19 vaccine leading to a dramatic reduction in hospitalisation and mortality, the study was closed in September 2022, before reaching the power needed to ~~conclude its end points~~ determine the effect of ruxolitinib in the treatment of COVID-19 pneumonia in patients requiring hospital admission.

~~Despite being underpowered to make a definitive conclusion, numerically lower odds (OR: 0.63) of progressing to severe COVID-19 pneumonia (modified WHO grade ≥ 5) were observed with ruxolitinib compared to SOC. However, the 95% CI is wide (0.25 to 1.57) and crosses 1 indicating uncertainty and requires cautious interpretation. In contrast, the odds of progression were numerically higher (OR 1.19) with fostamatinib compared to SOC. Although, again, results were imprecise with a wide 95% CI (0.51 to 2.76).~~